# Adaptation dynamics between copy-number and point mutations

**Isabella Tomanek[†], Călin C Guet***

Institute of Science and Technology Austria, Klosterneuburg, Austria

**Abstract** Together, copy-number and point mutations form the basis for most evolutionary novelty, through the process of gene duplication and divergence. While a plethora of genomic data reveals the long-term fate of diverging coding sequences and their *cis*-regulatory elements, little is known about the early dynamics around the duplication event itself. In microorganisms, selection for increased gene expression often drives the expansion of gene copy-number mutations, which serves as a crude adaptation, prior to divergence through refining point mutations. Using a simple synthetic genetic reporter system that can distinguish between copy-number and point mutations, we study their early and transient adaptive dynamics in real time in *Escherichia coli*. We find two qualitatively different routes of adaptation, depending on the level of functional improvement needed. In conditions of high gene expression demand, the two mutation types occur as a combination. However, under low gene expression demand, copy-number and point mutations are mutually exclusive; here, owing to their higher frequency, adaptation is dominated by copy-number mutations, in a process we term amplification hindrance. Ultimately, due to high reversal rates and pleiotropic cost, copy-number mutations may not only serve as a crude and transient adaptation, but also constrain sequence divergence over evolutionary time scales.

***For correspondence:**
calin@ist.ac.at

**Present address:** †Department of Biology, University of Oxford, Oxford, United Kingdom

**Competing interest:** The authors declare that no competing interests exist.

## Editor's evaluation

This is an important paper that proposes a novel evolutionary mechanism by which copy-number mutations can slow down the accumulation of point mutations in populations evolving in certain environments. The authors use an evolution experiment in bacteria equipped with a clever reporter system to provide convincing evidence that this mechanism indeed operates. This paper will be of broad interest to readers in evolutionary biology and related fields.

## Introduction

Adaptive evolution proceeds by selection acting on mutations, which are often implicitly equated with point mutations, that is, changes to a single nucleotide in the DNA sequence. However, nature is full of different types of bigger-scale mutations, such as mutations to the copy-number of genomic regions ranging from only a few base pairs up to half a bacterial chromosome (*Anderson and Roth, 1977*; *Darmon and Leach, 2014*). The specific properties of mutations, such as their rate of formation and reversal, might influence the evolutionary dynamics in major ways, but are rarely considered.

In bacteria, which are our focus, the duplication of genes or genomic regions occurs orders of magnitude more frequently than point mutations, ranging from $10^{-6}$ up to $10^{-2}$ per cell per generation (*Roth, 1988*; *Drake et al., 1998*; *Andersson and Hughes, 2009*; *Elez et al., 2010*; *Reams and Roth, 2015*). Moreover, while duplications can form via different mechanisms, they all are genetically unstable (*Andersson and Hughes, 2009*); the repeated stretch of DNA sequence is prone to *recA*-dependent homologous recombination. At rates between $10^{-3}$ and $10^{-1}$ per cell per generation duplications will reverse to the single copy (deletion) or duplicate further (amplification) (*Roth, 1988*; *Andersson and Hughes, 2009*; *Pettersson et al., 2009*; *Reams and Roth, 2015*; *Tomanek et al.,*

2020). Amplification of a gene or genomic region will, to a first approximation, increase its expression by means of elevated gene dosage (*Elde et al., 2012*; *Gruber et al., 2012*; *Näsvall et al., 2012*; *Yona et al., 2015*; *Steinrueck and Guet, 2017*; *Belikova et al., 2020*; *Todd and Selmecki, 2020*). Not surprisingly, due to their high rate of formation, gene amplifications are adaptive in situations where a rapid increase in gene expression is needed: resistance to antibiotics, pesticides or drugs via over-expression of resistance determinants (*Prody et al., 1989*; *Albertson, 2006*; *Bass and Field, 2011*; *Nicoloff et al., 2019*), immune evasion (*Belikova et al., 2020*), or novel metabolic capabilities through increased expression of spurious enzymatic side activities (*Blount et al., 2020*; *Richts et al., 2021*). Due to their high intrinsic rate of deletion, often combined with significant fitness cost (*Bergthorsson et al., 2007*; *Pettersson et al., 2009*; *Reams et al., 2010*), copy-number mutations not only differ from point mutations in their frequency of occurrence, but also in the nature of their reversibility.

Together, copy-number and point mutations are responsible for the evolution of most functional novelty of genes through the process of duplication and divergence of existing genes (*Ohno, 1970*; *Kacser and Beeby, 1984*; *Conant and Wolfe, 2008*; *Andersson et al., 2015*). Owing to the dynamic nature of gene duplication formation and reversal, the interplay between copy-number and point mutations may lead to complex evolutionary dynamics around the time point of origin of a new gene duplication event. However, so far most attention has been focused on understanding the long-lasting process of how duplicate gene pairs diverge by accumulating point mutations (*Lynch and Conery, 2000*; *Teufel et al., 2015*; *Friedlander et al., 2017*), while we know little about the potentially short-lived initial duplication event itself (*Innan and Kondrashov, 2010*). On one hand, this bias is due to significant technical challenges in studying transient copy-number variation experimentally (*Andersson and Hughes, 2009*; *Lauer and Gresham, 2019*; *Belikova et al., 2020*; *Tomanek et al., 2020*), and on the other hand, research has focused on the plethora of long-term evolutionary data that document the sequence divergence of paralogs, as 'attention is shifted to where the data are' (*Kondrashov, 2012*).

In bacteria adaptive amplification, that is, amplification as a response to selection as opposed to neutral duplication and divergence, is considered the default mode of paralog evolution (*Andersson and Hughes, 2009*; *Treangen and Rocha, 2011*; *Copley, 2020*) and has been conceptualized in the innovation-amplification-divergence (IAD) model (*Bergthorsson et al., 2007*), which was later validated by evolution experiments (*Elde et al., 2012*; *Näsvall et al., 2012*). The IAD model posits that selection for a novel enzymatic activity leads to adaptive gene amplification that increases expression of an existing enzyme if it exhibits low levels of a beneficial secondary enzymatic activity (also referred to as promiscuous functions; *Aharoni et al., 2005*; *Tawfik, 2010*; *Copley, 2017*). Eventually, protein sequences diverge as point mutations improve the secondary enzymatic function: a new protein function is born from an existing one. After the new (improved) function is present, superfluous additional gene copies will be lost due to their cost and high rate of reversibility, leaving only the copies of the two (ancestral and evolved) paralogs (*Bergthorsson et al., 2007*; *Reams et al., 2010*; *Elde et al., 2012*; *Näsvall et al., 2012*).

Similarly, adaptive amplification can precede the divergence of promoter sequences under selection favouring increased gene expression (*Steinrueck and Guet, 2017*). Thus, gene amplifications serve as a fast adaptation which can later be replaced by point mutations either within the coding region of a gene, increasing a cryptic enzymatic activity, or in its non-coding promoter region, increasing its expression (*Elde et al., 2012*; *Näsvall et al., 2012*; *Yona et al., 2015*; *Steinrueck and Guet, 2017*).

Since elevated numbers of gene copies provide an increased target for point mutations to occur (*San Millan et al., 2017*), it has been suggested that copy-number mutations speed up the process of divergence (*Andersson and Hughes, 2009*). However, if both, copy-number and point mutations, are adaptive (*Gruber et al., 2012*), they also have the potential to interact epistatically or due to clonal interference (*Gerrish and Lenski, 1998*). This interaction could result in unexpected evolutionary dynamics due to the different rates of formation and reversal of the two different mutation types.

To fill the knowledge gap that exists at around 'time zero' of the duplication-divergence process (*Innan and Kondrashov, 2010*), we designed a synthetic genetic system with which we can monitor, in real time, arising copy-number and point mutations in evolving populations of *Escherichia coli*. Importantly, while our results are also relevant to the divergence of paralogous protein sequences, here we study the process of divergence in a model gene promoter. Our genetic reporter system allows us to

phenotypically distinguish between copy-number and point mutations, by specifically selecting for the increased expression of an existing but barely expressed gene. With our system at hand, we set out to test whether adaptive copy-number mutations facilitate or hinder adaptation by point mutation.

## Results

The motivation for this work was sparked by an evolution experiment conducted in *E. coli* at a locus exhibiting high rates of gene amplification (*Steinrueck and Guet, 2017*), which failed to produce any evolved clones with point mutations and thus lead us to hypothesize that copy-number mutations may interfere with the evolution by point mutations under certain conditions.

### An experimental system that distinguishes copy-number and point mutations

To study the interplay between copy-number and point mutations during adaptation, we follow the fate of a barely expressed gene during its evolution towards higher expression. Our experimental system consists of an intact endogenous *galK* gene of *E. coli* that harbours a random promoter sequence (P0) that replaces its endogenous promoter. By growing *E. coli* in the presence of the sugar galactose, we are selecting for increased *galK* expression. Adaptation to selection for increased expression can happen by two different, non-mutually exclusives ways: through increased copy-number (duplication or amplification) or through point mutations in the P0 promoter region of *galK* (divergence) (*Tomanek et al., 2020*).

Importantly, our genetic reporter system allows us to distinguish between the two mutation types. *GalK is* part of a chromosomal reporter gene cassette and is transcriptionally fused to a *yfp* gene (*Figure 1A*). Hence, any increases in *galK* expression – be it by copy-number or point mutations – can be detected as increases in YFP expression. However, only mutations to the copy-number of the entire *galK* locus lead to an additional increase in the expression of an independently transcribed *cfp* gene downstream of *galK-yfp* (*Steinrueck and Guet, 2017*; *Tomanek et al., 2020*; *Figure 1A*, *Figure 1—figure supplement 1A–C*). Hence, increases in *yfp* alone indicate the divergence of the *galK* promoter sequence P0 by point mutations, while increases of both fluorophores indicate copy-number mutations of the whole locus. Finally, clones with increased *yfp* but without point mutations in P0 would indicate the presence of a *trans*-acting mutation at a different locus on the chromosome or a rare amplification event occurring independent of the repeated IS elements and excluding CFP (*Steinrueck and Guet, 2017*; *Tomanek et al., 2020*). Moreover, while in principle possible, an adaptive mutation in the coding sequence of *galK* itself is extremely unlikely to be selected under our experimental conditions given that growth is limited only by expression of the endogenous and fully functional galactokinase enzyme.

### Different substrate levels result in different enzyme expression demands

Our experimental environment consists of liquid minimal medium containing amino acids as a basic carbon and energy source, such that cells can grow even in the absence of *galK* expression (*Figure 1B* – grey line). Adding galactose to this basic medium renders *galK* expression highly beneficial. To characterize the relation between fitness and *galK* expression, we engineered a construct where the expression of *galK* is induced by the addition of arabinose. Growth rate increased along with *galK* expression and saturated at a certain expression level, which depended on the galactose medium used (*Figure 1B*). Thus, our system allows studying adaptation in environments with different gene expression demands: low concentrations of galactose demand a low level of *galK* expression (and increasing expression above this level does not add any extra benefit), while high concentrations of galactose demand a higher level of *galK* expression to obtain maximum growth rate. In other words, our experimental system allows selecting for different levels of improvement of a biological function (in our case increased *galK* expression) by growing cells in different galactose concentrations.

### Evolution of galK expression in IS+ and IS- strains

Given the vast range of duplication rates observed at different chromosomal loci in bacteria (*Roth, 1988*; *Andersson and Hughes, 2009*; *Elez et al., 2010*; *Reams and Roth, 2015*), our objective was

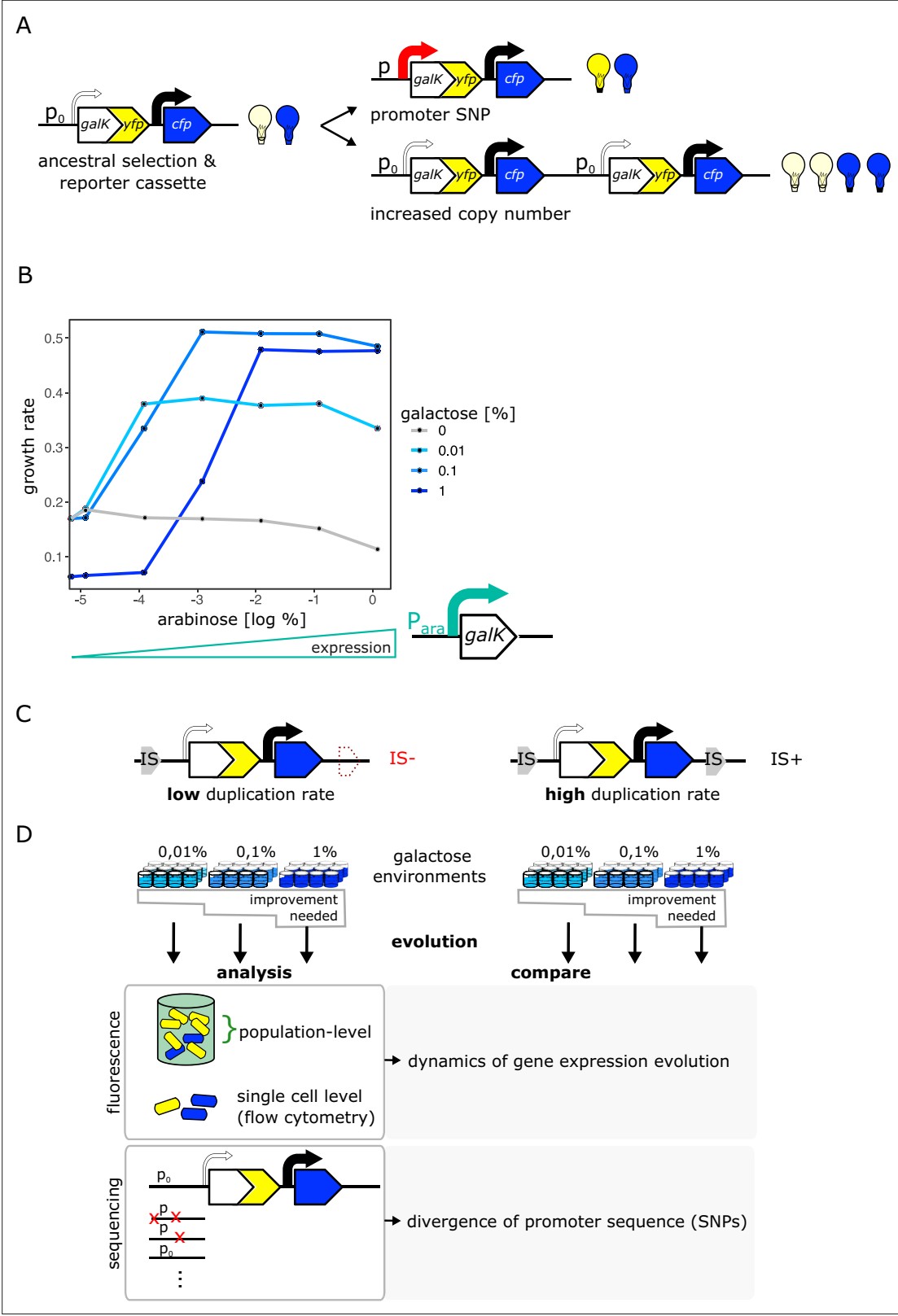

**Figure 1.** An experimental system to study gene duplication and divergence in strains with different duplication rates. (**A**) Cartoon of chromosomal selection and reporter cassette. The galK-yfp gene fusion does not have a functional promoter, but instead a random sequence, P0 (thin arrow), drives very low levels of baseline gene expression. Cfp expression is driven by a constitutive promoter (black arrow). Light bulbs symbolize fluorescence. Two fundamentally different kinds of adaptive mutations are shown on the right: (i) point mutations in P0 lead to increases in GalK-YFP while CFP remains

*Figure 1 continued on next page*

*Figure 1 continued*

at ancestral single-copy levels (top), (ii) mutations to the copy-number of the whole reporter cassette will increase both YFP and CFP expression (bottom). (**B**) Growth rate (as a proxy for fitness) as a function of different induction levels of galK expression in four different concentrations of galactose. Expression of a synthetic $p_{ara}$-galK cassette (schematic below the figure) is induced by the addition of arabinose. Growth rate increases along with increasing galK expression, but it plateaus at different values for different gene expression levels depending on galactose concentration (low, intermediate, and high gene expression demand). (**C–D**) Experimental layout. The adaptive dynamics and sequence divergence in P0 is compared between two otherwise isogenic strains (IS- and IS+) that differ in their rate of forming duplications. For IS- the second endogenous copy of IS1C located 12 kb downstream of the selection and reporter cassette has been deleted (**C**). Ninety-six replicate populations of each strain are evolved in three different levels of galactose, which select for increasing levels of gene expression improvement for 12 days, respectively. Throughout, fluorescence is analysed in bulk and on a single-cell level to analyse evolutionary dynamics, and relevant clones are sequenced (**D**).

The online version of this article includes the following source data and figure supplement(s) for figure 1:

**Source data 1.** Contains an R script along with optical density measurments to plot *Figure 1B*.

**Figure supplement 1.** An experimental system to study gene duplication and divergence in strains with different duplication rates.

**Figure supplement 1—source data 1.** Contains an R script along with qPCR and fluorescence intensity data to plot *Figure 1—figure supplement 1*.

to experimentally manipulate the ability of *galK* to form duplications and study its effect on evolutionary dynamics. A common way to manipulate the duplication rate is by deleting the *recA* gene involved in homologous recombination (*Goldberg and Mekalanos, 1986*; *Reams et al., 2010*; *Dhar et al., 2014*). However, given its role in DNA repair, comparing *recA* and *ΔrecA* strains will be strongly influenced by the growth defects that such a mutation entails. In order to not have to consider pleiotropic effects caused by a difference in the genome-wide duplication rate, we instead compare two identical strains whose difference in duplication rate is restricted to a single genomic locus. To this end, we take advantage of a chromosomal location that is characterized by high rates of duplication and amplification due to homologous recombination occurring between two endogenous identical insertion sequences (IS) elements that flank this specific locus (*Steinrueck and Guet, 2017*; *Tomanek et al., 2020*). By deleting one copy of IS*1*, we generated two otherwise isogenic strains of *E. coli* that differ solely by the presence of one IS*1* element approximately 10 kb downstream of *galK* (*Figure 1C*), and are thus predicted to show strong differences in their rates of duplication formation at this locus. In the following, we will refer to these strains as IS+ and IS-.

To understand how the duplication rate affects adaptive dynamics, we conducted an evolution experiment with 96 replicate populations of the IS+ and IS- strains (*Figure 1D*). Growing these populations in minimal medium containing only amino acids (control) or supplemented with three different galactose concentrations enabled us to follow adaptation to different gene expression demands (levels of selective pressure) (*Figure 2A*). Daily measurements of population fluorescence prior to dilution (1:820) allowed us to monitor population phenotypes roughly every 10 generations over 12 days.

The evolution experiment confirmed that the two strains differ strongly in their rate of copy-number mutations of the *galK* locus. The strain lacking one of the flanking IS*1* elements (IS-) showed a drastic reduction in the ability to undergo *galK* amplification. In contrast to the IS+ strain, very few IS- populations evolved increased CFP expression (*Figure 2A* – red traces). Interestingly, in the IS+ strain, the number of populations amplified by the end of the experiment depended on the environment. At least twice as many populations were amplified in the low (0.01%) galactose environment compared to the other two environments (68, 19, and 34 populations for low, intermediate, and high galactose, respectively) (*Figure 2—figure supplement 1A*). Not only the number of amplified populations, but also the maximum CFP fluorescence attained by IS+ populations differed significantly between the low (0.01%) and higher (0.1% and 1%) galactose environments (*Figure 2—figure supplement 1B*). Populations, which evolved increases in CFP fluorescence, did so within 2 days and maintained this level relatively stably for the duration of the experiment. (See *Figure 2—figure supplement 2A* for an independent evolution experiment confirming the environment-dependent patterns of amplification.) The observed difference in the number of *galK* copies is consistent with the observation that the three environments select for different levels of increasing gene expression ('levels of improvement') (*Figure 1B*) and confirms that amplifications are an efficient way of tuning gene expression (*Tomanek et al., 2020*).

We then asked whether other differences in the nature of adaptive mutations exist between the three different environments. To get a coarse-grained overview, we plotted the YFP fluorescence of evolving populations as a proxy for *galK* expression against their CFP fluorescence as a proxy for

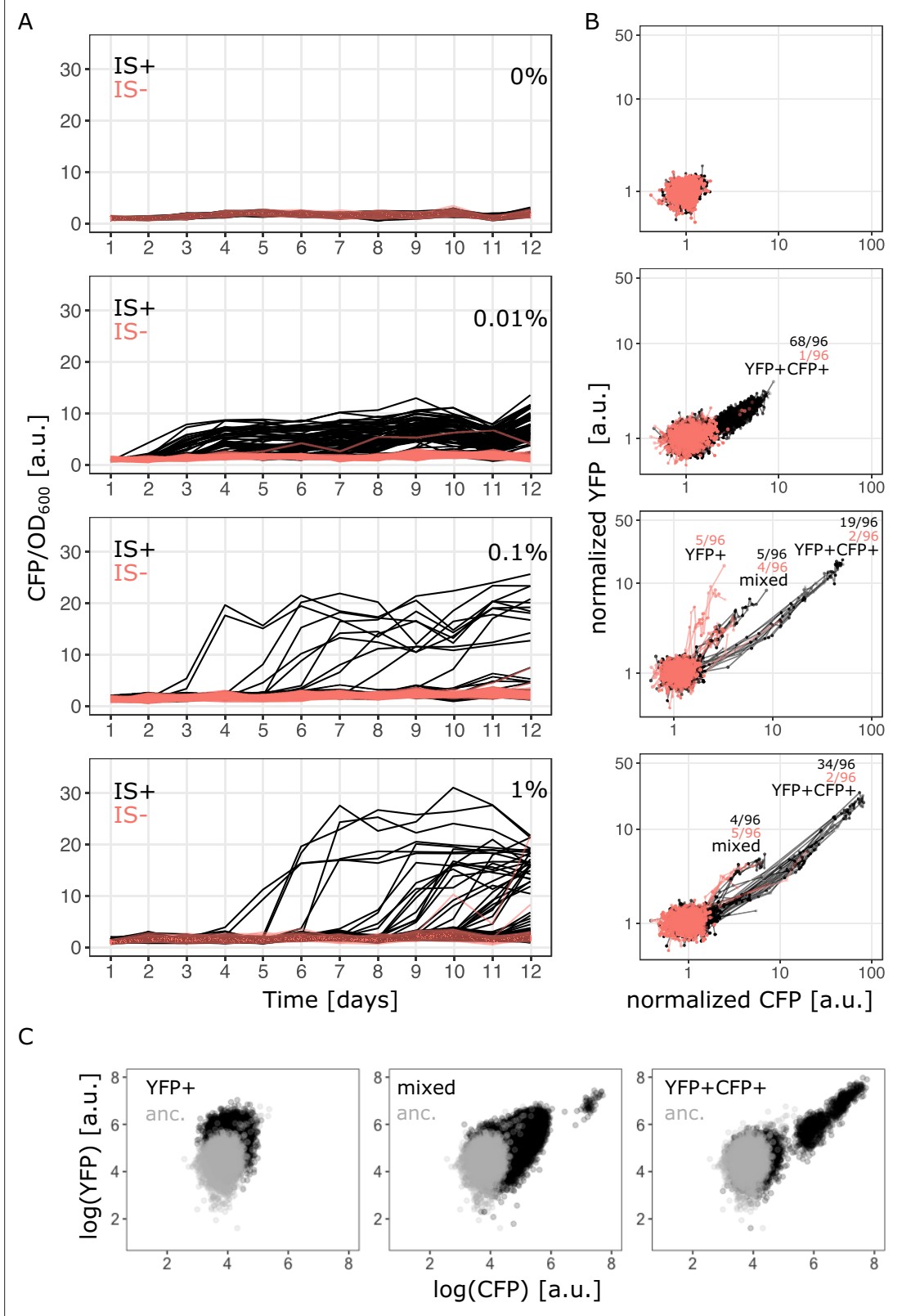

**Figure 2.** Evolutionary dynamics depend on galactose concentration and duplication rate. (**A**) Daily measurements of normalized CFP fluorescence as a proxy for gene copy-number of 96 populations of IS+ (black) and IS- (red) strains growing in three different galactose concentrations (% indicated in the plot), respectively, as well as 33 replicates of IS+ and IS- strain, respectively, growing in the absence of galactose (control, black). (**B**) Logarithmic plots for an overview of fold changes in YFP and CFP fluorescence of populations from (**A**) (YFP and CFP were normalized to the mean fluorescence of ancestral

*Figure 2 continued on next page*

*Figure 2 continued*

populations (anc.) evolved in 0% galactose [top panel]). Lines connect measurements of each population. Populations' fluorescence phenotypes occupy three different areas: increased YFP only (YFP+), increased CFP and YFP (YFP+CFP+ , i.e. amplified) and increased CFP with an additional elevation in YFP above the YFP+CFP+ fraction (mixed). The number of populations for IS- (red) and IS+ (black) in the respective fractions are indicated (see *Figure 1—figure supplement 1A* and *Figure 3A–B*). (**C**) Representative flow cytometry plots showing single-cell YFP and CFP fluorescence for populations from the YFP+ (left), mixed (middle), and YFP+CFP+ (right) fraction (indicated in panel **B**), respectively.

The online version of this article includes the following source data and figure supplement(s) for figure 2:

**Source data 1.** Contains an R script along with optical density and fluorescence data to plot *Figure 2A-B*.

**Figure supplement 1.** Number of amplified populations and their copy-number depends on the gene expression demand of the environment.

**Figure supplement 2.** Evolutionary dynamics depend on galactose concentration.

**Figure supplement 2—source data 1.** Contains an R script along with optical density measurments to plot *Figure 2—figure supplement 2B*.

**Figure supplement 3.** YFP-only amplifications occur in IS- populations evolved in 0.1% galactose.

**Figure supplement 3—source data 1.** Contains an R script along with qPCR data to plot *Figure 2—figure supplement 3B*.

*galK* copy-number for all time points (*Figure 2B*). The YFP-CFP plot shows that evolving populations exhibit qualitatively different distributions of fluorescence levels in the three different environments, indicating that adaptation has followed different trajectories.

In the absence of galactose, populations retain their ancestral fluorescence phenotype. In the lowest galactose concentration (0.01%), data points show a correlated increase between YFP and CFP fluorescence indicative of gene copy-number mutations ('YFP+CFP+' in *Figure 2B*). In the intermediate galactose concentration (0.1%) 5/96 IS- populations exhibit increased YFP fluorescence with ancestral (single-copy) CFP fluorescence indicative of promoter mutants, ('YFP+' fraction in *Figure 2B*; *Figure 2—figure supplement 3A*). However, sequencing the P0 region upstream of *galK* of these evolved clones from populations with strongly increased YFP fluorescence ('YFP+' fraction in *Figure 2B*) showed that they harboured an ancestral P0 sequence (*Figure 2—figure supplement 3A*). We hypothesized that the YFP+ populations carried an amplification extending into *galK-yfp*, yet excluding *cfp*. Quantitative real-time PCR confirmed our suspicion (*Figure 2—figure supplement 3B*). As the IS- strain cannot undergo the frequent duplication via the two flanking IS elements, it cannot access a major adaptive route available to the IS+ strain. Thus, its adaptation follows an alternative trajectory, which occurs through a repeat-independent lower-frequency duplication with junctions between *yfp* and *cfp* (*Figure 2—figure supplement 3C*).

While increased CFP still reliably reports on increased copy-number, the *yfp*-only amplification hijacks our ability to unambiguously infer ancestral copy-number from ancestral CFP fluorescence alone. As increasing CFP itself bears no adaptive benefit, populations with increased CFP must carry amplifications that also include *galK*. In contrast, ancestral copy-number can only be confirmed by qPCR. The fact that some populations carry IS-independent *yfp*-only amplifications implies that our system of fluorescence reporters will yield a slight underestimate of the number of amplified populations both in the IS+ and IS- strain. However, we were ultimately interested in the divergence of promoter sequences, and going forward relied on sequencing to unambiguously determine the presence of adaptive promoter mutations.

In the high (1%) and intermediate (0.1%) galactose environment, data points occupy an additional space ('mixed fraction' in *Figure 2B*) between the other two fractions, where both YFP and CFP are increased, but the YFP increase is larger than in the YFP+CFP+ fraction. Based on these population-level data, we hypothesized that this phenotypic space is occupied either by a population of double mutants carrying a combination of point and copy-number mutations, or by populations consisting of cells with only promoter mutations and cells with only copy-number mutations (i.e. the two mutations being mutually exclusive). Knowing the single-cell phenotype is therefore crucial for distinguishing between the two cases. Importantly, single-cell fluorescence (using FACS) recapitulated the population measurements with the YFP-CFP phenotype falling into three distinct fractions (*Figure 2C*).

## Copy-number and point mutations occur as a combination in the intermediate and high demand environment

To understand whether copy-number and point mutations are mutually exclusive or if they occur as a combination in the IS+ strain after evolution in intermediate (0.1%) and high (1%) galactose,

we determined the single-cell fluorescence of all mixed fraction populations using flow cytometry (*Figure 3A–B*). It is worth noting that after 12 days of evolution, cells with ancestral YFP and CFP fluorescence were still present in every single amplified population. While some populations consisted of a high fraction of cells with elevated CFP fluorescence, mutants did not yet spread to complete fixation in any of them, highlighting the fact that our experiments are capturing the transient adaptive dynamics.

Flow cytometry results showed that IS+ populations of the mixed fraction from intermediate (0.1%) galactose (*Figure 3A*) consisted of a single type of mutant with increased YFP/CFP fluorescence relative to the ancestral values (*Figure 3C*). If instead a population consisted of two mutually exclusive mutants, we would expect cells to fall into two distinct phenotypic clusters, one with only increased YFP (corresponding to the 'YFP+' fraction) and one with only amplifications (corresponding to the 'YFP+CFP+' fraction). Moreover, YFP fluorescence of the mixed fraction cells was greater than YFP for pure amplification mutants, which falls along the diagonal axis (*Figure 2C* – right panel), again indicating a combination of copy-number and promoter mutations. To confirm the presence of combination mutants, we randomly picked three populations of the mixed fraction. Sequencing revealed that within these populations, only amplified clones, but not clones with single-copy *cfp* harboured an SNP (–30T>A) in P0 (*Figure 3E*).

Similar to intermediate galactose, IS+ populations from the high (1%) galactose mixed fraction (*Figure 3B*) harboured cells with the combination mutation phenotype and, in addition, cells with pure amplifications (*Figure 3D*). Taken together, these data indicate that copy-number and point mutations can occur as a combination in environments with sufficiently high gene expression demand.

## Copy-number and point mutations are mutually exclusive in the low demand environment

After finding combination mutants in the high galactose environments, we analysed the single-cell fluorescence of all IS+ populations from the low (0.01%) galactose environment. Surprisingly, and in contrast to the intermediate and high galactose environments, in low galactose adaptive amplification of IS+ populations happened more rapidly with the majority of populations (68/96) showing increases in CFP fluorescence during the course of the experiment (*Figure 4A* – left top and bottom panel, *Figure 4—figure supplement 1A–B*). Notably, cells of those few populations that did not follow this general trend (*Figure 4A* – right top and bottom panel) showed an increase in YFP without a concomitant increase in CFP. As this small increase in YFP was not visible in the initial population measurements of liquid cultures (*Figure 2B*), we turned to patching populations onto LB agar, a potentially more sensitive method, which alleviates changes in fluorescence related to growth rate. Imaging populations confirmed the increase in YFP for all populations with elevated YFP in single-cell measurements (*Figure 4—figure supplement 2A*). We first examined population B1 with clearly increased YFP more carefully by re-streaking it on LB agar (*Figure 4C*). Consistent with flow cytometry results (*Figure 4B*), we found colonies with three different fluorescence phenotypes: ancestral, increased YFP ('YFP+'), and a small subpopulation with both, increased YFP and CFP (amplified). Sequencing of the amplified colony type confirmed it to be a bona fide amplification without additional promoter SNPs. Sequencing of the YFP+ colony type uncovered two adaptive SNPs in P0 (–30T>A and –37C>T), which were identical to a previously identified promoter mutation 'H5' (*Figure 2—figure supplement 2B*; *Steinrueck and Guet, 2017*; *Tomanek et al., 2020*).

As we failed to find combination mutants (i.e. a mixed fraction) in population measurements from the low galactose environment (*Figure 2B*), we used agar patches from four different time points of the evolution experiment (*Figure 4—figure supplement 2A*) to screen IS+ populations more comprehensively (*Figure 4D*). Re-streaking, sequencing and flow cytometry analysis revealed that all populations with elevated YFP and ancestral CFP (*Figure 4D* – red triangles) harboured either only promoter mutants or a mixed population of a few amplified cells and a majority of promoter mutants (*Table 1*). As opposed to high and intermediate galactose, we did not find a single population with combination mutants in low galactose. Moreover, the fact that mutations were mutually exclusive within populations was also reflected when we analysed their fate over time. Quantitative analysis of the fluorescence intensity of patched populations (*Figure 4D*) confirmed that populations with a significant fraction of promoter mutants (i.e. visibly YFP+ on the agar patch) did not become amplified later in the experiment. As a single exception, population F6 gained the YFP+ phenotype early, but

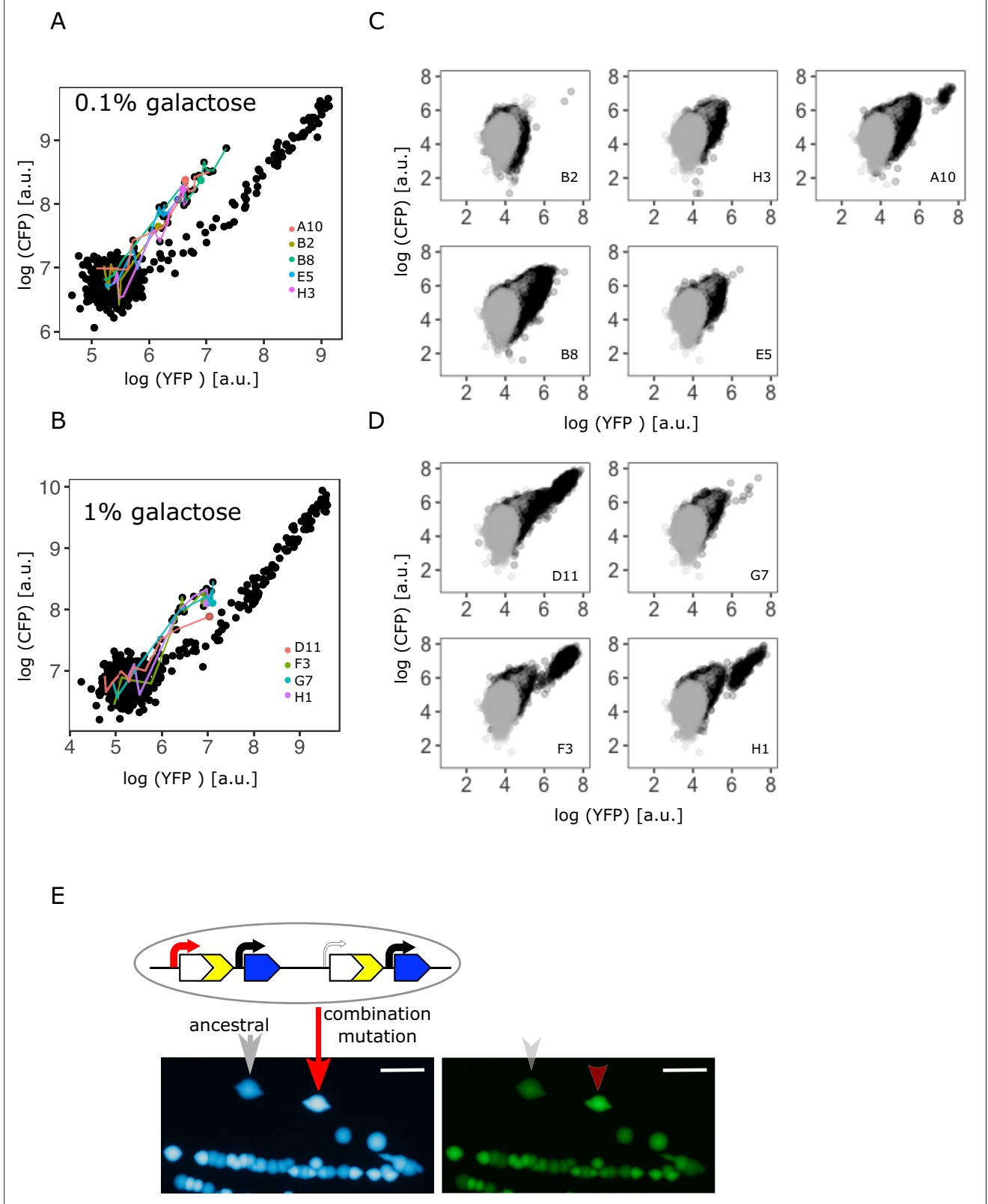

**Figure 3.** Confirming the presence of a combination of copy-number and point mutations in intermediate and high galactose. (**A–B**) Log plot of YFP and CFP fluorescence of all 96 IS+ populations during evolution in 0.1% (**A**) and 1% (**B**) galactose (black points), respectively. Data replotted from *Figure 2B* for an overview of population fluorescence of all mixed fraction populations (coloured points). Time points of measurements are indicated by the degree of shading. (**C–D**) Single-cell fluorescence phenotypes as measured by flow cytometry of all mixed fraction populations identified in

*Figure 3 continued on next page*

Figure 3 continued

(**A–B**) after 12 days of evolution, respectively, indicate the presence of combination mutations (an increase of both YFP and CFP within a single cell as opposed to a mixed population of cells with either an increase in YFP or an increase in CFP, compare to **Figure 2C**). (**E**) Sanger sequencing of individual colonies allows to determine the genotype of an evolved clone of any fluorescence phenotype. Images of CFP (left) and YFP (right) fluorescence of individual colonies from a representative IS+ population (A10) streaked onto LB agar after having evolved in 0.1% galactose for 12 days. Sanger sequencing of the P0 sequence revealed a T>A point mutation in an amplified (red arrow) but not an ancestral colony (grey arrow). Scalebars: 1cm.

The online version of this article includes the following source data for figure 3:

**Source data 1.** Contains an alignment of sequencing data for **Figure 3E**.

became dominated by gene amplifications by the end of the experiment (**Figure 4D** – right panel, blue triangle). Nevertheless, also in this case, copy-number and point mutations did not occur in the same genetic background. Conversely, all YFP+ populations evolved exclusively from those with ancestral phenotype; no single amplified population gained a functional promoter within the time frame of the experiment (**Figure 4D**).

The complete absence of combination mutants in the low demand environment is consistent with the fact that only a modest increase in *galK* expression is necessary to reach maximal fitness (**Figure 1B**). Thus, while a combination of amplification and promoter point mutation evolves in response to selection for a strong increase in *galK* expression (intermediate and high demand environments), either mutation alone might provide a sufficient increase in gene expression to allow for maximal growth in the low demand environment. This would mean that the fitness benefit of either mutation does not add up when combined. In other words, we hypothesize that negative epistasis precludes the evolution of combination mutants in the low demand environment. Alternatively, the lack of combination mutants could be explained by clonal interference between competing adaptive amplifications and point mutations (a possibility we discuss in the last section of Results).

## An increased fraction of adaptive promoter mutations is found in IS- populations evolved in the low demand environment

If copy-number mutations are more frequent than point mutations and their combination does not spread to observable frequencies in the low demand environment, we would expect divergence to proceed more slowly as compared to an intermediate or high demand environment.

To directly test this hypothesis, we estimated the level of divergence between all of the IS+ and IS- populations evolved in the low demand (0.01% galactose) environment. We pooled all 96 populations into pools of 32 and quantified the fraction of SNPs in P0 previously known to be adaptive (**Tomanek et al., 2020**). To do so, we subjected PCR amplicons of the pooled populations to next-generation sequencing (**Figure 5A**, **Figure 5—figure supplement 1A**). We designed our sequencing experiment such that we were able to analyse 39 bp upstream and downstream of the *galK* start codon. We calculated the fraction of sequence reads carrying either one or both most frequently observed adaptive SNPs at position –30 and –37 upstream of the *galK* start codon (**Table 1**). As a control, we also compared the fraction of SNPs within the *galK* gene of the IS+ and IS- evolved under different galactose conditions. In our experimental system, galactose selection is not expected to lead to adaptive mutations anywhere in the coding region of *galK*, as the enzyme itself is fully functional despite lacking a functional promoter sequence. Comparing the fraction of reads with SNPs (i.e. reads with a single SNP in *galK* divided by the number of reads with ancestral *galK*) allowed us to compare across samples with different absolute numbers of sequencing reads (**Figure 5—figure supplement 1A**). Consistent with our expectation, the fraction of sequencing reads with a single SNP at any position in *galK* was similar in populations evolved in different galactose concentrations and in the control populations evolved in the absence of galactose (**Figure 5A–B**).

We then compared the fraction of reads with the two adaptive SNPs in P0 previously known to confer increased *galK* expression (**Figure 5A**). While the fraction of reads carrying SNPs in *galK* is similar in all media, SNPs in P0 were more frequent in media containing galactose than in the control (**Figure 5A** – left and right panels) in agreement with strains adapting to galactose selection. Intriguingly, in low galactose, we found a higher fraction of reads carrying both adaptive single SNPs (–30T>A and –37C>T) in IS- populations than in the IS+ populations. This is consistent with our hypothesis that the more frequent amplification mutants effectively out-compete point mutations in the low demand environment.

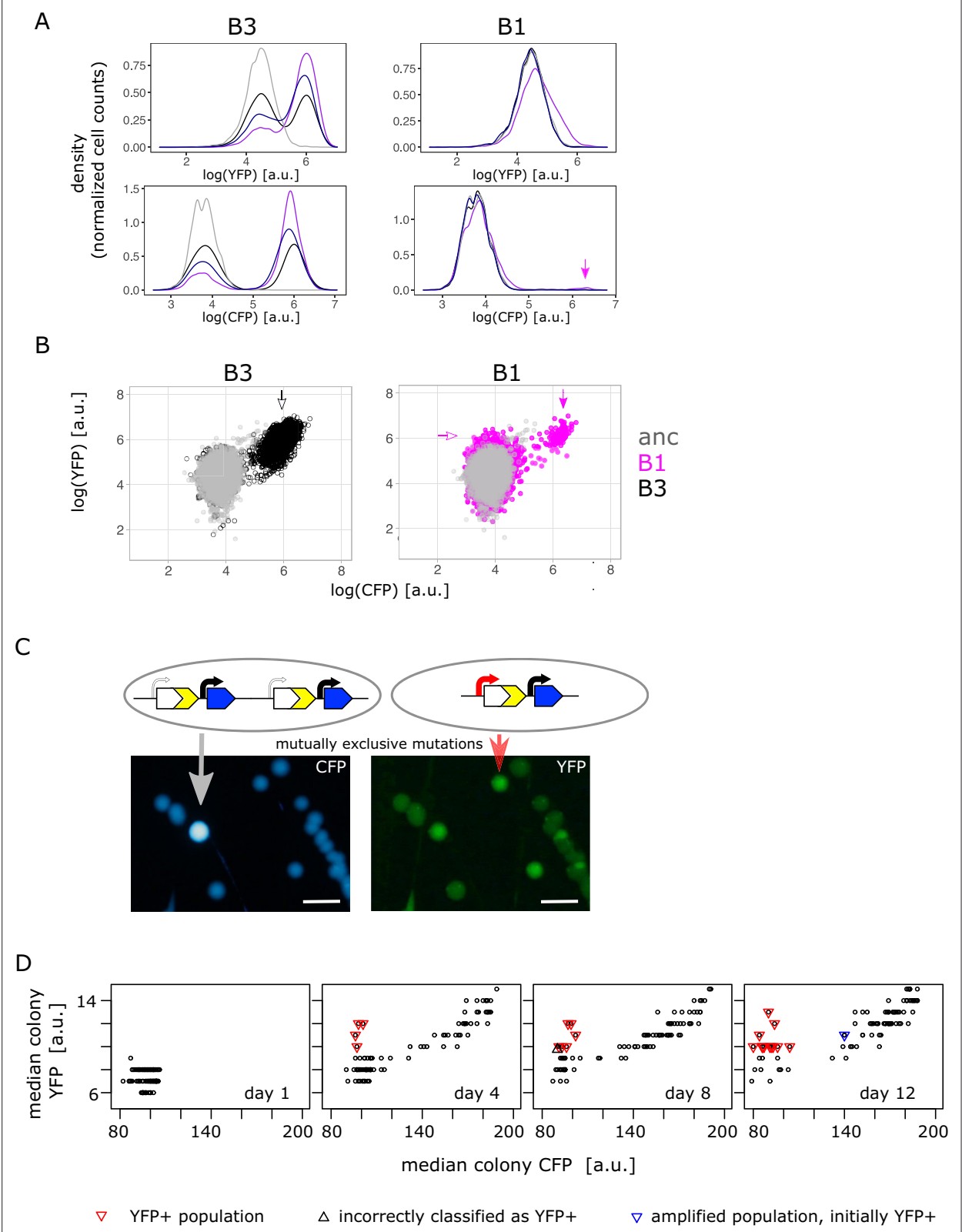

**Figure 4.** Confirming the presence of mutually exclusive mutations in low galactose. (**A**) Representative flow cytometry density plot showing YFP fluorescence (upper left and right panel) and CFP fluorescence (lower left and right panel) of IS+ populations B3 (left panels) and B1 (right panels) over time (grey – ancestral, black – day 4, dark blue – day 8, purple – day 12). The small YFP+CFP+ subpopulation is indicated by a magenta arrow (see corresponding arrow in **B** – right panel). (**B**) YFP versus CFP plot of populations B3 (left panel, black) and B1 (right panel, magenta) at day 12 together

*Figure 4 continued on next page*

*Figure 4 continued*

with an ancestral population (grey) in order to better visualize the two distinct subpopulations in B1 (magenta arrows indicate YFP+ and YFP+CFP+ subpopulation, respectively). Data is replotted from A in order to visualize subpopulations. (**C**) Images of CFP (left) and YFP (right) fluorescence of individual colonies from IS+ population B1 (shown in **B**) streaked onto LB agar after 12 days of evolution in 0.01% galactose. The population consists of amplified colonies with increased CFP and YFP fluorescence (grey arrows) and single-copy colonies with a promoter mutation (red arrows). Scalebars: 1cm. (**D**) Quantitative analysis of patched populations indicates that promoter mutants (YFP+) evolve only in single-copy backgrounds. YFP-CFP plot of median colony fluorescence intensity of populations patched onto agar (as shown in **B**) on days 1, 4, 8, and 12 of evolution in 0.01% galactose. Populations were classified as YFP+ if their YFP but not CFP fluorescence intensity values exceeded ancestral fluorescence (red triangles, confirmed by flow cytometry). In all these populations, the YFP+ phenotype evolved from an ancestral phenotype. Blue triangle represents an amplified population, which was classified as YFP+ in the previous time point (flow cytometry showed that this population became dominated by copy-number mutations later). Black triangle marks population incorrectly classified as YFP+ (ancestral fluorescence according to flow cytometry). See also *Table 1*.

The online version of this article includes the following source data and figure supplement(s) for figure 4:

**Source data 1.** Contains an R script along with colony fluorescence intensity data over time to plot *Figure 4D*.

**Figure supplement 1.** Adaptation to the low galactose environment is dominated by gene amplification.

**Figure supplement 2.** Monitoring population fluorescence under neutral conditions with respect to *galK* expression reveals small increases in YFP fluorescence in the absence of amplification.

We are here using the fraction of sequencing reads ('alleles') with adaptive SNPs divided by the number of ancestral reads as a simple metric of divergence. However, this normalization leads to an underestimation of SNPs if they occur in an amplified background. For instance, a SNP within a cell with four P0-*galK* copies, where one carries an SNP, counts less than a cell with one copy of P0-*galK* carrying one SNP. The rationale for using the fraction of adaptive alleles as our metric of divergence as opposed to the alternative, which is the number of SNPs per cell, is twofold: First, the methodology used here does not allow comparing absolute read counts between samples. Second, and more importantly, due to the random nature of deletion mutations, a single SNP in an amplified array of four copies has a one in four chance of being retained as a lasting divergent copy in the process of

**Table 1.** Sequencing and phenotypic analysis of all YFP+IS+ populations evolved in 0.01% galactose (*Figure 4D* – red triangles).

Increase in fluorescence relative to ancestral (anc) phenotype indicated by YFP+ and CFP+. Results shown for day 12 populations unless otherwise noted (d4, d8).

| Population | Seq (all YFP+) | Flow cytometry phenotype | Agar streak | Comment |
|---|---|---|---|---|
| A6 | –30T>A | YFP+, v. few CFP+ (mixed populations) | YFP+, few CFP+ | |
| B1 | –30T>A, –37C>T ("mutation H5") | YFP+, CFP+ (mixed populations) | Few YFP+, few CFP+, mixed pop | |
| B2 | –30T>A | YFP+ | YFP+, v. few CFP+ | |
| C1 | –30T>A | YFP+ (d12) | YFP+, v. few CFP+ | |
| C9 | – | Ancestral YFP (d8), only CFP+(d12) | Few CFP+ | Incorrectly classified as YFP+ (*Figure 4D* – grey triangle) |
| D2 | –30T>A | YFP+ (d12) | YFP+ only | |
| D9 | anc | YFP+ (d8, d12) | YFP+ only | |
| E10 | –30T>A | YFP+ (d12) | YFP+ only | |
| F6 | – | YFP+ (d4), CFP+(d12) | CFP+ subpopulation | YFP+ at d8, then amplified population (*Figure 5D* – blue triangle) |
| F10 | –30T>A | YFP+, CFP+, anc (mixed populations) | YFP+, CFP+, mixed pop | qPCR confirmed |
| G1 | –30T>A | YFP+, v. few CFP+ (d12) | YFP+, v. few CFP+ | |
| G12 | –30T>A | YFP+ (d8) | YFP+, no CFP+ (d12) | FACS CFP+ carry over |

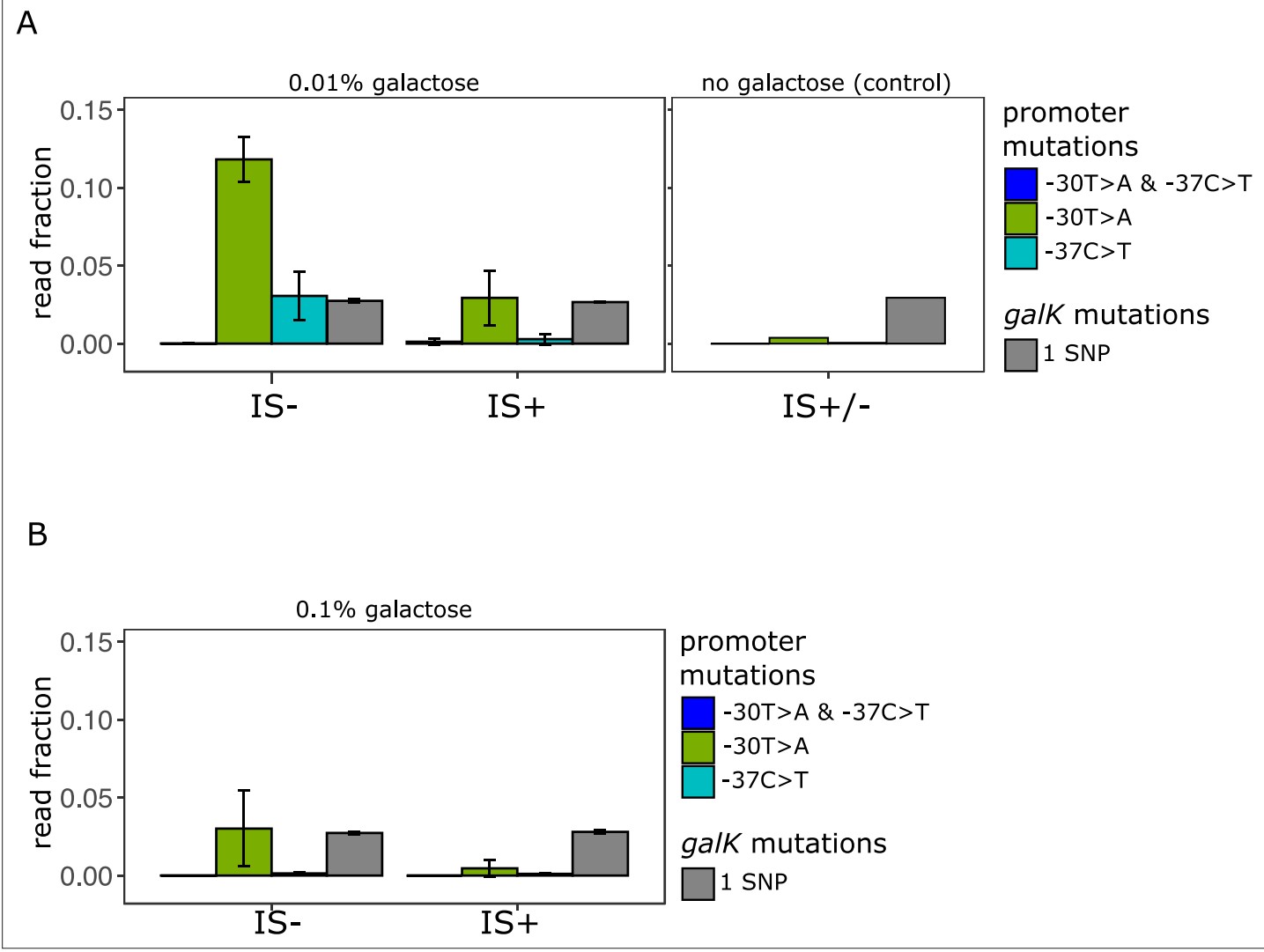

**Figure 5.** Amplicon deep sequencing of P0 in pooled evolved populations. (**A**) (Left panel) Number of reads carrying a P0 sequence with two adaptive SNPs 30 and 37 bp upstream of *galK*, respectively ('T>A + C>T' in blue) or its respective single SNPs ('T>A' in green, 'C>T' in cyan). Values are normalized to the number of reads with ancestral P0 for IS- and IS+ populations evolved in 0.01% galactose. The mean fraction of reads with any single SNP in *galK* is shown as a control (grey). Error bars represent the standard deviation of three replicates, consisting each of 32 pooled evolved populations. (Right panel) Read fractions of the same respective SNPs shown for a pool of all 96 IS+ and IS- populations evolved in the absence of galactose. (**B**) Mean read fractions as in (**A**) shown for three replicates of each 32 pooled populations evolved in intermediate (0.1%) galactose.

The online version of this article includes the following figure supplement(s) for figure 5:

**Figure supplement 1.** Total number of sequencing reads for all replicates.

amplification and divergence. Hence, the dilution of SNPs by additional amplified copies is not simply a counting artefact, but reflects a biological reality relevant to the very process that we are studying. Therefore, we conclude that in the low demand environment a strain which cannot adapt by gene amplification exhibits a higher level of divergence than a strain which frequently adapts by gene amplification.

## Evolutionary dynamics between mutation types differ for different initial random promoter sequences

Given the paucity of point mutations that we observed for the evolution of the random P0 sequence (either a combination of –30T>A and –37C>T or each SNP alone), we wondered whether a greater variety of mutations could be obtained when using a different random promoter sequence as a

starting point for evolution. Therefore, we repeated our evolution experiment in the intermediate (0.1%) galactose environment with three additional random promoter sequences (P0-1, P0-2, P0-3).

After 10 days of evolution, only two out of the four random P0 sequences evolved increased *galK-yfp* expression (*Figure 6A*). This is roughly consistent with the fact that approximately 60–80% of random sequences are one point mutation away from a functional constitutive promoter (*Yona et al., 2018*; *Lagator et al., 2022*). Interestingly, P0-1 and P0-3 did not gain any gene duplications or amplifications. At first glance, this drastic difference in gene amplification was unexpected, since the IS+ strains only differ in their P0 sequence, and not in their gene duplication rate. However, random sequences have different abilities to recruit RNA-polymerase, and as a result, different baseline expression levels (*Yona et al., 2018*; *Lagator et al., 2022*). Given that a plateau exists in the expression growth relation for low levels of expression (*Figure 1B*), the initial expression level conferred by P0-1 and P0-3 might be too low to yield a selective benefit upon gene duplication alone. According to this hypothesis, these random (non-)promoters are not only two (or more) point mutations away from a beneficial sequence, but also two (or more) copy-number mutations.

## Copy-number and point mutations are mutually exclusive in the intermediate demand environment for P0-2

For P0, the evolution experiment in intermediate galactose reproduced our previous findings, namely a YFP+CFP+ (amplified) and a mixed (amplified with increased YFP) fraction for IS+ populations and a YFP+ fraction for IS- populations (compare *Figure 6A* with *Figure 2B*), which corresponds to an amplification of YFP, but not CFP (*Table 2*).

For P0-2, the evolutionary dynamics differed from P0. In the IS+ strain, almost every single population evolved amplifications within the first 2 days of the evolution experiment (*Figure 6B*, *Figure 6—figure supplement 1A*). Moreover, only two fractions are visible in the YFP-CFP plots of P0-2. The first fraction is occupied by YFP+ populations carrying a single copy of *cfp*. The second fraction along the diagonal between YFP and CFP is occupied by amplified populations (YFP+CFP+). Moreover, it is shifted towards higher values of YFP/CFP relative to values found for P0 (*Figure 6—figure supplement 1B*), suggesting that P0-2 exhibits a higher baseline expression level than all the other three random promoter sequences. In contrast to the population-level measurements, single-cell measurements were not sufficiently sensitive to corroborate any difference in leaky expression amongst the four random promoter sequences (*Figure 6—figure supplement 1C*). However, in line with the observed evolutionary dynamics, P0 and even more so P0-2 confers a significant growth advantage over the other two promoters (*Figure 6—figure supplement 1D*). As mentioned above, this suggests that the observed growth advantage of P0-2 populations can explain their rapid amplification dynamics. In agreement with the evolution experiments with P0, the YFP+CFP+ (amplification) fraction is also strongly reduced in the IS- strain for P0-2.

Intriguingly, with the majority (88/96) of P0-2 IS+ population amplified, six P0-2 IS+ populations that failed to evolve amplifications show an increase in YFP/CFP early in the evolution experiment (*Figure 6B* – left panel, *Figure 6—figure supplement 1A*). This result combined with the idea that P0-2 exhibits a relatively high baseline expression level and the absence of a mixed fraction for P0-2 (*Figure 6A*) suggests that increases in gene expression evolve *either* via gene amplification *or* via point mutation. In other words, because initial *galK* expression is high in P0-2, a small improvement (either amplification or a promoter mutation) is sufficient to reach the required gene expression demand. Thus, the adaptive trajectory of P0-2 in intermediate galactose resembles that of P0 in low galactose as both environments select only for a modest improvement in *galK* expression.

In contrast to the IS+ strain, where only six populations showed increased YFP/CFP fluorescence that emerged only within the first 3 days of evolution, populations of the IS- strain were evolving increased YFP/CFP fluorescence throughout the experiment (*Figure 6B* – right panel). We were curious whether the increase in YFP/CFP in both, IS+ and IS- populations, was due to promoter mutations. Sequencing of randomly picked evolved clones revealed that in the majority (4/6 for IS+, 11/21 for IS-) of clones with increased YFP/CFP indeed harboured a mutation in P0-2, including an SNP, a 12 and a 13 bp deletion (*Table 2*; *Figure 6C*). Importantly, colonies of the same populations but with ancestral fluorescence harboured ancestral P0-2 sequences (*Table 1*), indicating that the observed mutations (*Table 2*) are causal for the increased YFP expression. While finding the causal mutations for the remaining evolved clones with increased YFP but ancestral P0-2 (*Figure 6C*) lies outside the scope

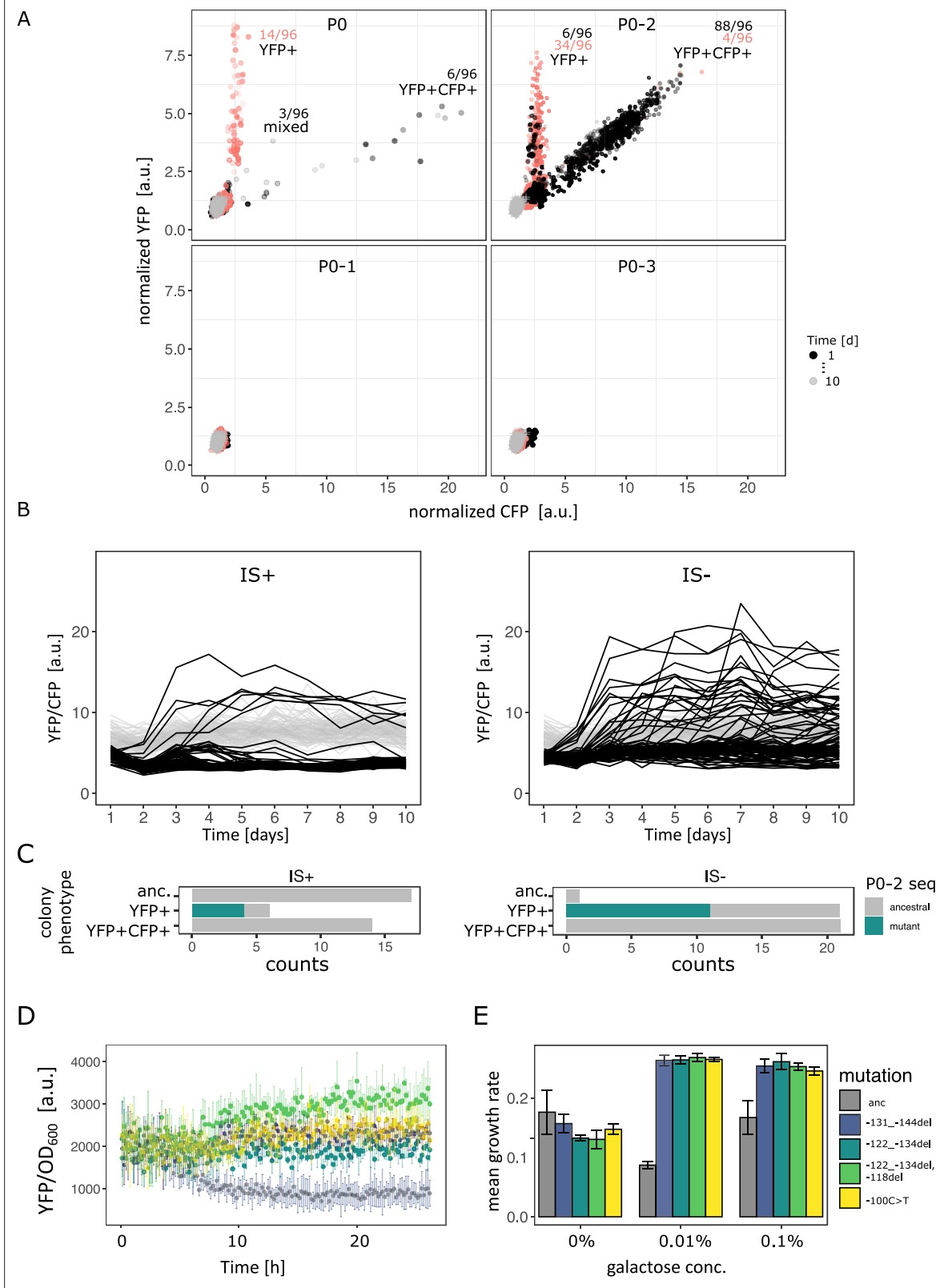

**Figure 6.** Evolutionary dynamics for different random P0 sequences in 0.1% galactose. (**A**) YFP versus CFP fluorescence normalized to the ancestral value of 96 populations of IS+ (black) and IS- (red) strain each harbouring a different random sequence upstream of galK ('P0', 'P0-1', 'P0-2', 'P0-3') grown in 0.1% galactose and without galactose (grey lines, control), respectively. Time points are indicated by the degree of shading. The number of populations for IS- (red) and IS+ (black) in the respective fractions are indicated. (**B**) YFP/CFP fluorescence to visualize increases in galK-YFP expression

*Figure 6 continued on next page*

*Figure 6 continued*

not caused by copy-number increases plotted for the duration of the evolution experiment for P0-2 populations of IS+ (left panel) and IS- (right panel). Here, gene amplifications (see *Figure 6—figure supplement 1A*) are visible as slight decrease in YFP/CFP relative to the 0% galactose control (grey), putative promoter mutations are visible as an increase in YFP/CFP. (**C**) Distribution of P0-2 mutants in IS+ and IS- populations after 12 days of evolution in 0.1% galactose. Mutations in P0-2 are exclusively found in populations with increased YFP and ancestral CFP fluorescence (YFP+). IS+ clones from all six YFP+ populations were sequenced, while IS- clones from a random subsample of 21 YFP+ populations were sequenced. (**D**) Mean normalized YFP fluorescence of reconstituted P0-2 mutants and the P0-2 ancestor strain (grey) grown in control medium (0% galactose). (**E**) Mean growth rate of reconstituted P0-2 mutants and the ancestor strain (grey) in 0.01% galactose, 0.1% galactose, and control medium (0% galactose). Error bars represent the standard deviation of four replicates.

The online version of this article includes the following source data and figure supplement(s) for figure 6:

**Source data 1.** Contains an R script along with optical density and fluorescence intensity measurments to plot *Figure 6A-B*.

**Source data 2.** Contains an R script along with optical density and fluorescence intensity measurments to plot *Figure 6D-E*.

**Figure supplement 1.** Rapid amplification of IS+ populations with P02.

of the current work, we speculate that they may occur further upstream of P0-2 or could be acting in trans such as mutations in the transcription factor *rho* (*Steinrueck and Guet, 2017*).

To confirm that the 12 bp deletion mutation, the 13 bp deletion mutation and the SNP were in fact adaptive, we reconstituted these mutations into the ancestral P0-2 strain, where they conferred increased YFP expression (*Figure 6D*) resulting in increased growth in medium supplemented with galactose (*Figure 6E*). The finding that the promoter mutations were responsible for increased *galK-yfp* expression was corroborated by the fact that these mutations occurred exclusively in populations with increased YFP but ancestral CFP, and were completely absent in amplified (YFP+CFP+) and ancestral colonies from a random set of 14 IS+ populations (*Figure 6C*). It is worth noting that mutations observed in P0-2 were more diverse than those observed in P0 (seven different mutations including indels, an IS insertion and an SNP in P0-2 versus three different SNPs in P0 – compare *Tables 1 and 2*). Thus, amplification can interfere with divergence not only by point mutations but also by small insertions and deletions.

Taken together, the facts that (i) the majority of IS+ populations become rapidly amplified, (ii) with few promoter mutations arising exclusively in the first day in non-amplified populations (mutations are mutually exclusive), and (iii) many more promoter mutations occur in IS- populations throughout the evolution experiment strongly suggest that negative epistasis between frequent copy-number mutation and point mutations hinder fixation of the latter.

**Table 2.** Mutations of P0-2 underlying increased YFP fluorescence in IS+ and IS- populations evolved in 0.1% galactose.

| IS+ clones | | IS- clones | |
|---|---|---|---|
| P02-A11 | −131_−144del | P02-A7 | −100C>T |
| P02-B10 | −122_−134del | P02-H12 | −100C>T |
| P02-F4 | −100C>T | P02-C3 | −100C>T |
| P02-F4 | −100C>T, poor quality read | P02-H9 | −122_−134del |
| | | P02-F2 | −122_−134del |
| | | P02-D1 | −100C>T |
| | | P02-E2 | −100C>T |
| | | P02-A1 | Bigger band, maps to *insD1* coding sequence |
| | | P02-E5 | −41del |
| | | P02-C5 | 201 bp deletion leaving 20 bp of P02 |
| | | P02-H5 | 201 bp deletion leaving 20 bp of P02 |
| | | | (seven different kinds of mutations) |

## Amplification hinders divergence by point mutations in the low demand environment

The experimental results we presented this far suggest that the evolutionary dynamics of duplication/amplification and divergence depend on the level of gene expression increase selected for (*Figure 7*). In both environments, promoter point mutations evolve at a low rate in a single-copy background. However, if rates of copy-number mutation are high, evolutionary dynamics are dominated by amplification. Irrespective of the environment, this amplification increases the mutational target size for rarer adaptive point mutations to occur. However, only if a strong increase in *galK* expression is selected for (high demand environment), the beneficial effects of both types of mutation add up, and we observe a combination of amplifications and point mutations to occur, in agreement with the IAD model (*Bergthorsson et al., 2007*; *Näsvall et al., 2012*; *Andersson et al., 2015*; *Figure 7A*).

The IAD model assumes that amplification and point mutations only occur in the same genetic background. However, whether the two different types of mutation fix consecutively in the *same* genetic background or in different competing clones depends on the effective population size and the respective mutation rates (*Gerrish and Lenski, 1998*). High rates of duplication and amplification may cause clonal interference between competing mutants, slowing down the fixation of either. Moreover, there needs to be sufficient selective benefit ('demand') for two consecutive selective sweeps to occur. If, however, only a modest level of gene expression increase is selected for (low demand environment) (*Figure 1B*), a single mutational event may be sufficient to provide it. Therefore, adaptation is dominated by the more frequent type of mutation, namely copy-number mutation. In other words, amplifications effectively hinder divergence in the low demand environment due to their negative epistatic interaction with point mutations. Thus, in a process, which we term amplification hindrance, the high rate of amplification results in evolutionary dynamics that slow down divergence via two different non-mutually exclusive mechanisms: clonal interference and negative epistasis.

However, in our experiments mutation rates can be assumed to be equal across environments. Moreover, in the absence of *galK* expression (i.e. for the ancestral strain) population sizes are similar across different galactose concentrations (*Figure 7—figure supplement 1A*). Hence, clonal interference is an unlikely explanation for the absence of combination mutants in the low galactose environment. However, there is a difference in the degree to which strains that harbour amplifications fulfil the necessary gene expression demand posed by the environment they have evolved in. Strains with amplifications evolved in the high and intermediate galactose environment grow slower and to lower densities than a strain with a strong constitutive promoter. In contrast, in the low galactose concentration strains with amplifications evolved in this environment exhibit both yield and growth rate comparable to that of the promoter mutant strain (*Figure 7—figure supplement 1B–D*).

These results suggest that gaining additional promoter point mutations on top of an amplification would only be beneficial in the higher galactose concentrations, but yield little or no fitness benefit in the low galactose environment. Therefore, under the experimental conditions presented here, gene expression demand – and hence negative epistasis – plays a major role in amplification hindrance.

## Discussion

In this study, we investigated the interaction dynamics between two different types of mutations, adaptive copy-number and point mutations. While the process of gene duplication and divergence per se has been intensely studied since the pioneering work of Ohno more than half a century ago, no experiments have scrutinized the early phase of this process, where transient evolutionary changes may prevail. So far, the few existing experimental studies simply introduced mutations a priori without studying their formation dynamics (*Dhar et al., 2014*), while in silico studies used genomics to query the 'archaeological' results of millions of years of sequence evolution (*Innan and Kondrashov, 2010*).

Here, we used experimental evolution to investigate how the early adaptive dynamics of diverging promoter sequences is influenced by the rate of copy-number mutations as well as the level of expression increase selected for. We found that the spectrum of adaptive mutations differed drastically between environments selecting for different levels of expression of the same gene (*Figures 1B, 3A and 6A*). Combination mutants carrying both, copy-number and promoter point mutations, only evolved under conditions selecting for big increases in the levels of *galK* expression. In contrast, selection for only a modest increase in *galK* expression lead to populations

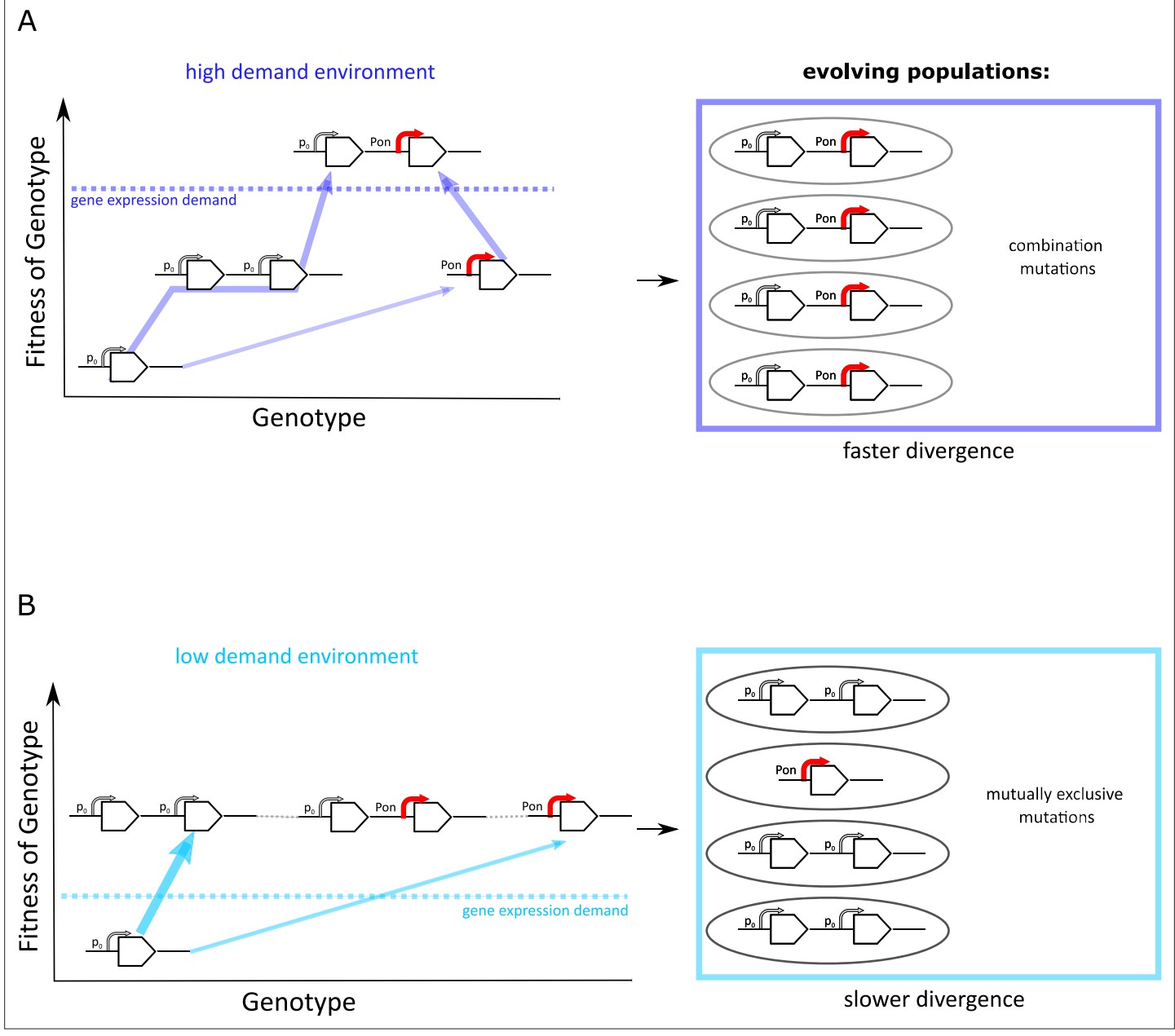

**Figure 7.** Frequent copy-number mutation can hinder adaptation by point mutations. Genotype-fitness map ('fitness landscape') illustrating the difference between adaptive trajectories of a high demand (**A**) and low demand (**B**) environment, which differ solely by the increase in gene expression they select for. The dashed line indicates the level of gene expression sufficient to reach maximal growth rate ('fitness') (see also *Figure 1B*). Right panels show the experimentally observed genotypes for each environment. (**A**) For an environment selecting for a large increase in gene expression (high demand), more than one adaptive mutation is necessary to reach maximal fitness. If copy-number mutations are frequent (as in the IS+ strain), adaptation by amplification is most likely (bold arrow). Alternatively, at a lower frequency, adaptation occurs via a point mutation in the promoter sequence (thin arrow). Due to an increased mutational target size, cells with gene amplfications are more likely to gain a beneficial point mutation than cells with a single copy of galK. Alternatively, rare promoter mutants can become amplified, in either case leading to the combination mutant observed in experiments. (**B**) For an environment selecting for only a modest increase in gene expression (low demand), maximal growth rate is attained either by gene amplification (more frequent, bold arrow) or by point mutations (less frequent, thin arrow). Therefore, combination mutants do not provide an additional fitness benefit and would only increase in frequency due to drift (horizontal faint dashed lines), not selection. Combination mutants are not observed in the experiment (right panel).

The online version of this article includes the following figure supplement(s) for figure 7:

**Figure supplement 1.** Amplification hindrance is consistent with negative epistasis under conditions of low gene expression demand.

adapting by either gene amplifications or point mutations in their random promoter sequence, but not both simultaneously. Moreover, if amplification occurred early in the experiment, the random promoter sequence P0 did not diverge within the timespan of the experiment (*Figure 4D*). This phenomenon was even more pronounced for a second random promoter sequence, P0-2 (*Figure 6B–C*).

Moreover, comparing the number of point mutations between strains that differ solely in the rate of undergoing copy-number mutations in the *galK* locus, we found that under a low demand environment, a strain with a high duplication rate (IS+) diverged more slowly compared to a strain with low duplication rate (IS-).

Taken together, our results suggest that frequent gene amplification hinders the fixation of adaptive point mutations due to most likely negative epistasis between these two different mutation types. While epistatic interactions can occur with any two adaptive mutations, copy-number mutations are unique, in that they are orders of magnitude more frequent than point mutations in bacteria (*Roth, 1988*; *Drake et al., 1998*; *Andersson and Hughes, 2009*; *Elez et al., 2010*; *Reams and Roth, 2015*) and in eukaryotes (*Lynch et al., 2008*; *Lipinski et al., 2011*; *Schrider et al., 2013*; *Keith et al., 2016*). This large difference in rates means that a competition between point and copy-number mutations is heavily skewed in favour of the latter (*Figure 7B*).

Unlike the phenomenon of clonal interference (which occurs between any two beneficial mutations even if their adaptive benefits are additive) (*Gerrish and Lenski, 1998*), negative epistasis does not slow down adaptation per se, as adaptation is agnostic to whether point or copy-number mutations lead to an improved phenotype. However, negative epistasis slows down divergence as populations have reached the fitness peak with an alternative kind of adaptive mutation. Negative epistasis between point and copy-number mutations can be expected to occur in any selective condition, which requires only a relatively modest increase to a particular biological function, namely an increase in gene expression or enzyme activity by only a *few*-fold. Thus, amplification hindrance may not only be of general relevance for the evolution of gene expression in bacteria, but also for the evolution of promiscuous enzyme functions, which analogous to a barely expressed gene can be enhanced by either copy-number mutations or point mutations in the coding sequence.

While we found that amplification slows down divergence under conditions of negative epistasis, the consensus in the literature has been that copy-number mutations not only serve as a first step in the 'relay race of adaptation' (*Yona et al., 2015*), but that they also facilitate divergence, either indirectly by providing a first 'crude' adaptation to cope with a new environment until more refined adaptation occurs by point mutations, or directly by increasing the target size for point mutations (*Andersson and Hughes, 2009*; *Elde et al., 2012*; *Yona et al., 2015*; *Cone et al., 2017*; *Bayer et al., 2018*; *Lauer et al., 2018*; *Todd and Selmecki, 2020*). The intuitive idea that amplification speeds up divergence (*Andersson et al., 1998*) was originally developed as strong evidence against the adaptive mutagenesis hypothesis proposed by Cairns and others (*Cairns et al., 1988*; *Cairns and Foster, 1991*).

Based on it, various experimental studies interpreted observations of adaptation to dosage selection in the light of 'amplification as a facilitator of divergence' (*Song et al., 2009*; *Pränting and Andersson, 2011*; *Elde et al., 2012*; *Näsvall et al., 2012*; *Yona et al., 2012*; *Yona et al., 2015*; *Cone et al., 2017*; *Bayer et al., 2018*; *Lauer et al., 2018*; *Todd and Selmecki, 2020*). However, despite showing that adaptive amplification *precedes* divergence by point mutations, none of the studies provided a direct experimental test of the hypothesis that amplification *causes* increased rates of divergence. Experiments controlling for the rate of amplification were needed in order to dissect the ensuing evolutionary dynamics and establish causality.

All else being equal, more copies indeed mean more DNA targets for point mutations to occur (*San Millan et al., 2017*). However, as our experiments show, all else is not necessarily equal, and the evolutionary dynamics may differ strongly between an organism that can increase copy-number as an adaptation and an organism that cannot. Intriguingly, indications for more complex dynamics can be found in the existing literature (*Yona et al., 2012*; *Lauer et al., 2018*; *Richts et al., 2021*). One study showed that rapid adaptive gene amplification in yeast results in strong clonal interference between lineages (*Lauer et al., 2018*). A second study in yeast found that adaptation to an abrupt increase in temperature was dominated by rapid copy-number mutation, with SNPs occurring only much later (*Yona et al., 2012*; *Yona et al., 2015*). Lastly, an experimental evolution study in *Bacillus*, adaptation

was dominated by copy-number mutations and the authors noted the surprising lack of promoter mutations (*Richts et al., 2021*).

The transient dynamics of gene amplification allows tuning of gene expression on short evolutionary time scales in the absence of an evolved promoter (*Tomanek et al., 2020*). In principle, such transient evolutionary dynamics do not leave traces in the record of genomic sequence data on evolutionary time scales and as such, their detailed study may not seem warranted. This is especially true in the context of duplication and divergence of paralogs, which is studied because abundant genomic sequence data are available (*Kondrashov, 2012*). Our present study proved this intuition wrong, as we uncovered a potentially long-lasting effect resulting from the transient dynamics associated with copy-number mutations: if adaptation by amplification is the fastest and sufficient, other, less frequent, mutations may not have a chance to compete. While our evolution experiments were conducted under continuous selection, natural environments are often characterized by regimes of fluctuating selection. Due to the pleiotropic cost often associated with copy-number increases as well as their high rate of deletion, adaptive amplification returns to the ancestral single-copy state in the absence of selection (*Andersson and Hughes, 2009*; *Reams et al., 2010*). This means that once the selective benefit of the transient adaptation ceased, no change at the level of genomic DNA remains (*Roth, 1996*). Therefore, the idea that gene amplifications act as a transient 'regulatory state' rather than a mutation (*Roth, 1996*; *Tomanek et al., 2020*) can be extended by an implication found here, namely that amplifications could effectively act as buffer against long-lasting point mutations. In this view, amplification could repeatedly provide rapid adaptation to selection for increased gene expression, but collapse back to the single-copy ancestral state once selection has subsided and yet hinder sequence divergence each time it does so. Thus, on sufficiently long time scales, the transient dynamics that play out before the fixation of mutations may ultimately shape entire genomes (*Cvijović et al., 2018*).

Amplification hindrance is in agreement with the observation that gene duplication and divergence is not a dominant force in the expansion of protein families in bacteria (*Treangen and Rocha, 2011*; *Tria and Martin, 2021*). Consequently, in all situations where rapid amplification provides sufficient adaptation, amplification hindrance could work as a mutational force that – in addition to purifying selection – acts to conserve existing genes and their expression level. While purifying selection affects deleterious alleles only, counterintuitively, amplification hindrance prevents beneficial mutations from fixating.

## Methods

### Bacterial strain construction

To construct the IS- strain, we replaced the second copy of IS*1* downstream of the selection and reporter cassette in IT030 (*Tomanek et al., 2020*) with a kanamycin cassette using pSIM6-mediated recombineering (*Datta et al., 2006*). Recombinants were selected on 25 µg/ml kanamycin to ensure single-copy integration.

To generate the additional random promoters sequences P0-1, P0-2, and P0-3, we generated 189 nucleotides using the 'Random DNA sequence generator' (https://faculty.ucr.edu/~mmaduro/random.htm) with the same GC content as P0 (55%). We synthesized these three sequences as gBlocks (Integrated DNA Technology, BVBA, Leuven, Belgium) with attached XmaI and XhoI restriction sites, which we used to clone P0-1, P0-2, and P0-3 into plasmid pMS6* (*Tomanek et al., 2020*) by replacing P0. We used pMS6* with the respective P0 sequence as a template to amplify the selection and reporter cassette and integrate it into MS022 (IS+) and IT049 (IS-) as described previously (*Tomanek et al., 2020*).

>P0
ACCGGAAAGACGGGCTTCAAAGCAACCTGACCACGGTTGCGCGTCCGTATCAAGATCCTCTTAA
TAAGCCCCCGTCACTGTTGGTTGTAGAGCCCAGGACGGGTTGGCCAGATGTGCGACTATATCGC
TTAGTGGCTCTTGGGCCGCGGTGCGTTACCTTGCAGGAATTGAGGCCGTCCGTTAATTTCC.

>P0_1
GTAGGCCCGCACGCAAGACAAACTGCTGGGGAACCGCGTTTCCACGACCGGTGCACGATTTAAC
TTCGCCGACGTGACGACATTCCAGGCAGTGCCTCCGCCGCCGGACCCCCCTCGTGATCGGGTAG
CTGGGCATGCCCTTGTGAGATATAACGAGAGCCTGCCTGTCTAATGATCTCACGGCGAAAG.

>P0_2
TCGGGGGGACAGCAGCGGCTGCAGACATTATACCGCAACAACACCAAGGTGAGATAACTCCGTA
GTTGACTACGCGTCCCTCTAGGCCTTACTTGACCGGATACAGTGTCTTTGACACGTTTGTGGGC
TACAGCAATCACATCCAAGGCTGGCTATGCACGAAGCAACTCTTGGGTGTTAGAATGTTGA.

>P0_3
CCCCTGTATTTGGGATGCGGGTAGTAGATGAGCGCAGGGACTCCGAGGTCAAGTACACCACCCT
CTCGTAGGGGGCGTTCCAGATCACGTTACCACCATACCATTCGAGCATGGCACCATCTCCGCTG
TGCCCATCCTGGTAGTCATCATCCCTATCACGCTTTCGAGTGTCTGGTGGCGGATATCCCC.

## Reconstitution of P0-2 mutants in the ancestral strain

The reconstituted P0-2 mutant strains were obtained using pSIM6-mediated oligo recombineering (*Sawitzke et al., 2011*) of the ancestral strain and selecting recombinants on M9 0.1% galactose agar. The sequence of the oligonucleotides used is listed below. Successful recombinants were confirmed by Sanger sequencing of P0-2. Amongst the recombinants transformed with the –122_–134del construct, we also recovered one colony with higher YFP fluorescence intensity than the other recombinants. Sequencing showed an additional single deletion (–118del) in addition to the –122_–134del created by recombineering. Fluorescence and growth rate of the serendipitously obtained mutant is shown in *Figure 6D–E* along with the three intended mutants.

>A11 oligo (–131_–144del)
ACCGCAACAACACCAAGGTGAGATAACTCCGTAGTTGACTGGCCTTACTTGACCGGATACAGTG
TCTTTGACACGTTTGTGGG.

>H12 oligo (–100C>T)
CTAGGCCTTACTTGACCGGATACAGTGTCTTTGATACGTTTGTGGGCTACAGCAATCACATCCA
AGGCTG.

>F2 oligo (–122_–134del)
CAACACCAAGGTGAGATAACTCCGTAGTTGACTACGCGTCCCTTGACCGGATACAGTGTCTTTG
ACACGTTTGTGGGCTACAGCA.

## List of strains used

| Strain name | Genotype | Purpose | Source |
|---|---|---|---|
| MG1655 | F⁻ λ⁻ ilvG- rfb-50 rph-1 | Strain background for all evolution experiments | Lab collection |
| IT013-TCD | BW27784, JA23100::*galP*, *mglBAC*::FRT, *galK*::FRT, locus1::pBAD-*galK* | Strain with pBAD-*galK* for testing expression-growth relation | *Tomanek et al., 2020* |
| BW25142 | lacIq rrnB3 ΔlacZ4787 hsdR514 Δ(araBAD)567 Δ(rhaBAD)568 ΔphoBR580 rph-1 galU95 ΔendA9 uidA(ΔMluI)::pir-116 recA1 | Host for *pir* plasmid pMS6* | *Khlebnikov et al., 2001* |
| MS022 | MG1655, JA23100::*galP*, *mglBAC*::FRT, *galK*::FRT | IS+ background for ancestor strain construction | Lab collection |

*Continued on next page*

*Continued*

| Strain name | Genotype | Purpose | Source |
|---|---|---|---|
| IT030 | MS022 locus2::P0-RBS-*galK* -RBS-*yfp* -FRT-pR-*cfp* | IS+ ancestor strain | *Tomanek et al., 2020* |
| IT049 | MS022 deleted for IS*1*C | IS- background for ancestor strain construction | This study |
| IT049-P0 | IT049 locus2::P0-RBS-*galK* -RBS-*yfp* -FRT-pR-*cfp* | IS- ancestor strain P0 | This study |
| IT049-P0-1 | IT049 locus2::P0-1-RBS-*galK* -RBS-*yfp* -FRT-pR-*cfp* | IS- ancestor strain P0-1 | This study |
| IT049-P0-2 | IT049 locus2::P0-2-RBS-*galK* -RBS-*yfp* -FRT-pR-*cfp* | IS- ancestor strain P0-2 | This study |
| IT049-P0-3 | IT049 locus2::P0-3-RBS-*galK* -RBS-*yfp* -FRT-pR-*cfp* | IS- ancestor strain P0-3 | This study |
| MS022-P0 | MS022 locus2::P0-RBS-*galK* -RBS-*yfp* -FRT-pR-*cfp* | IS+ ancestor strain P0 | This study |
| MS022-P-01 | MS022 locus2::P0-1-RBS-*galK* -RBS-*yfp* -FRT-pR-*cfp* | IS+ ancestor strain P0-1 | This study |
| MS022-P0-2 | MS022 locus2::P0-2-RBS-*galK* -RBS-*yfp* -FRT-pR-*cfp* | IS+ ancestor strain P0-2 | This study |
| MS022-P0-3 | MS022 locus2::P0-3-RBS-*galK* -RBS-*yfp* -FRT-pR-*cfp* | IS+ ancestor strain P0-3 | This study |
| IT030-H5r | MS022 locus2::pconst-RBS-*galK* -RBS-*yfp* -FRT-pR-*cfp* | Strain with constitutive *galK* expression conferred by two SNPs in P0 | *Tomanek et al., 2020* |
| IT030-D8c | MS022 locus2::pconst-RBS-*galK* -RBS-*yfp* -FRT-pR-*cfp* | Strain with constitutive *galK* expression conferred by one SNP in P0 | *Tomanek et al., 2020* |

## List of primers used

| Name | Sequence | Purpose |
|---|---|---|
| E_flank_f | GCTGGAGCCACTTGTAGCC | cassette integration test locus 2, sequencing P0s |
| E_flank_r | TCCTTGCTGAATCATTTTGTTC | cassette integration test locus 2 |
| P0_check_Fw | GTGTGAGTGGCAGGGTAG | sequencing P0s (together with E_flank_f) |
| qPCR_*galK* _Fw | GCTACCCTGCCACTCACA | estimating *galK* copy number |
| qPCR_*galK* _Rv | CGCAGGGCAGAACGAAAC | estimating *galK* copy number |
| rbsB_qPCR_Fw | GGCACAAAAATTCTGCTGATTAA | qPCR control locus |
| rbsB_qPCR_Rv | GCAGCTCGATAACTTTGGC | qPCR control locus |
| P1_P0-1 | GCCTTAGTTGTAAGTGTCTACCATGTCCCCGAACAAGTGTTCACTATGTCTAGGCCCGCACGCAAGAC | integration of the selection and reporter cassette with P0-1 (Fw primer) |
| P1_P0-2 | GCCTTAGTTGTAAGTGTCTACCATGTCCCCGAACAAGTGTTCACTATGTCTCGGGGGGACAGCAGCG | integration of the selection and reporter cassette with P0-2 (Fw primer) |
| P1_P0-3 | GCCTTAGTTGTAAGTGTCTACCATGTCCCCGAACAAGTGTTCACTATGTCTGTATTTGGGATGCGGGTAGTAGA | integration of the selection and reporter cassette with P0-3 (Fw primer) |
| E_int_Rv | TCGGAAGGGAAGAGGGAGTGCGGGAAATTTAAGCTGGATCACATATTGCCGAGGCCTTATGCTAGCTTC | integration of the selection and reporter cassette (Rv primer) |

*Continued on next page*

*Continued*

| Name | Sequence | Purpose |
|------|----------|---------|
| E_int_Fw | GCCTTAGTTGTAAGTGTCTACCATGTC CCCGAACAAGTGTTCACTATGTCACCG GAAAGACGGGCTTC | integration of the selection and reporter cassette with P0 (Fw primer) |
| deep_seq_Fw | TCGTCGGCAGCGTCAGATGTGTATAAG AGACAGACGGGTTCTTATGCCTTAGTT | 1st step PCR for amplicon deep sequencing (with 5′nextera anchor for Illumina sequencing) |
| deep_seq_Rv | GTCTCGTGGGCTCGGAGATGTGTATAA GAGACAGGTGTGAGTGGCAGGGTAG | 1st step PCR for amplicon deep sequencing (with 5′nextera anchor for Illumina sequencing) |

## Culture conditions

Bacterial strains were grown at 37°C. All evolution experiments, as well as growth experiments with the purpose of measuring $OD_{600}$ and fluorescence, were conducted in M9 medium supplemented with 2 mM $MgSO_4$, 0.1 mM $CaCl_2$, 0.1% casaminoacids ('evolution medium'), and carbon source (galactose, glucose, or glycerol) at the concentration indicated in the respective figures (Sigma-Aldrich, St Louis, MO), with the exception of *Figure 2—figure supplement 2B*, where bacteria were grown in M9 medium without casaminoacids (carbon sources as indicated in the figure).

## Evolution experiments

Evolution experiments were inoculated with ancestral colonies of IS+ and IS- strains grown in 3 ml of LB medium over night, after two washing steps in M9 medium without carbon source (M9 buffer) and a dilution of 1:200.

Bacterial cultures were grown in 200 µl liquid evolution medium with the indicated galactose concentrations in clear flat-bottom 96-well plates and shaken in a Titramax plateshaker at 750 rpm (Heidolph, Schwabach, Germany), allowing for a total population size of ~$10^8$ colony forming units for the ancestral strain. Every day, populations were transferred to fresh plates using a VP408 pin replicator (V&P SCIENTIFIC, Inc, San Diego, CA) resulting in a dilution of ~1:820 (*Steinrueck and Guet, 2017*), corresponding to ~10 generations. Immediately after the transfer, growth and fluorescence measurements were performed in the overnight plates using a Biotek H1 plate reader (Biotek, Vinooski, Vermont). Thus, population phenotypes were measured every 10 generations.

## Growth rate measurements in liquid cultures

To measure the growth rate in a 2D gradient of arabinose and galactose (*Figure 1B*), an overnight culture of strain IT013 (*Tomanek et al., 2020*) grown in M9 supplemented with 1% glycerol and 0.1% casaminoacids was diluted 1:200 into 96-well plates containing 200 µl of M9 supplemented with 0.1% casaminoacids, with concentrations of galactose and the inducer arabinose as indicated in *Figure 1B*. For the full duration of the experiment, cultures were grown in the plate reader with continuous orbital shaking and $OD_{600}$ and fluorescence was measured in 10 min intervals.

Growth rate was calculated using a custom R script. Briefly, the script applies a linear model (base R function lm()) to a 20-datapoint sliding window of $log(OD_{600})$ as a function of time. The script then outputs the steepest slope (maximal growth rate) amongst all possible sliding windows (*Figure 1— source data 1*). The growth rates plotted in *Figure 6E* and *Figure 2—figure supplement 2B* were obtained in the same manner (see *Figure 6—source data 2* and *Figure 2—figure supplement 2— source data 1*), with strains and carbon sources as indicated in the respective figures.

## Flow cytometry experiments

Frozen evolved populations (–80°C, 15% glycerol) from day 4, day 8, or day 12 (as indicated in the figures) were pinned (1:820) into M9 buffer and put on ice until the measurement. Fluorescence was measured using a BD FACSCanto II system (BD Biosciences, San Jose, CA) equipped with FACSDiva software. CFP fluorescence was collected with a 450/50 nm bandpass filter by exciting with a 405 nm laser. YFP fluorescence was collected with a 510/50 bandpass filter by exciting with a 488 nm laser. The bacterial population was gated on the FSC and SSC signal resulting in approximately 6000 events analysed per sample, out of 10,000 recorded events.

## Quantitative real-time PCR

For qPCR, gDNA was isolated from overnight cultures grown in the respective evolution medium inoculated by single evolved colonies using Wizard Genomic DNA purification kit (Promega, Madison, WI).

We performed qPCR using Promega qPCR 2× Mastermix (Promega, Madison, WI) and a C1000 instrument (Bio-Rad, Hercules, CA). To quantify the copy-number of samples of an evolving population, we designed one primer pair within *galK* (target) and one primer within *rbsB* as a reference, which lies outside the amplified region. We compared the ratios of the target and the reference loci to the ratio of the same two loci in the single-copy control. Using dilution series of one of the gDNA extracts as template, we calculated the efficiency of primer pairs and quantified the copy-number of *galK* in each sample employing the Pfaffl method, which takes amplification efficiency into account (*Pfaffl, 2001*). qPCR was performed in three technical replicates.

## Measurement of colony fluorescence

Evolving populations were pinned onto LB agar supplemented with 1% charcoal and imaged using a macroscope setup (https://openwetware.org/wiki/Macroscope) (*Chait et al., 2010*). To obtain median colony YFP and CFP fluorescence intensity, a region of interest was determined using the ImageJ plugin 'Analyze Particles' (settings: 200px-infinity, 0.5–1.0 roundness) to identify colonies on 16-bit images with threshold adjusted according to the default value. The region of interest including all colonies was then used to measure intensity and plotted using a custom R script (*Figure 4—source data 1*).

## Amplicon deep sequencing of P0

Frozen samples of evolved populations were diluted 1:10 into 100 µl of LB and grown for 5 hr (37°C, shaking) to increase cell numbers prior to DNA extractions. Columns 1–4 (populations A1, B1, C1, …, F4, G4, H4), 5–8 (populations A5, B5, C5, …, F8, G8, H8), and 9–12 (populations A9, B9, C9, …, F12, G12, H12) of each 96-well plate were pooled prior to DNA extraction using Wizard Genomic DNA purification kit (Promega, Madison, WI). The P0 region including the beginning of *galK* was amplified for 25 PCR cycles using primers deep_seq_Fw and deep_seq_Rv carrying 5′ adaptors for Illumina sequencing. In parallel, PCRs were performed for 35 cycles to confirm bands on a gel. Illumina sequencing was carried out by Microsynth (Balgach, Switzerland).

We note that our amplicon libraries of P0 were contaminated with reads carrying the sequence of P0-2, which we had prepared for sequencing in parallel (*Figure 5—figure supplement 1*). We therefore excluded all reads of P0-2 for our analysis of P0 and do not report the result of the P0-2-specific samples as they could not be trusted.

Reads of P0 were analysed using a custom R script. Briefly, we defined four sequence motifs of each 39 bp length, which represented the ancestral P0 sequence and the same region with known adaptive SNPs (–30T>A, –37C>T or both). We calculated the fraction of reads with an evolved versus ancestral 39 bp motif in all samples, including those of control populations evolved in the absence of galactose. We also calculated the fraction of reads carrying a 39 bp ancestral *galK* sequence motif with any single single SNP versus those with the same 39 bp motif of the ancestral *galK* sequence.

## Acknowledgements

We are grateful to N Barton, F Kondrashov, M Lagator, M Pleska, R Roemhild, D Siekhaus, and G Tkacik for input on the manuscript and to K Tomasek for help with flow cytometry.

## Additional information

### Funding

No external funding was received for this work.

### Author contributions

Isabella Tomanek, Conceptualization, Data curation, Software, Formal analysis, Validation, Investigation, Visualization, Methodology, Writing – original draft, Project administration, Writing – review and editing; Călin C Guet, Conceptualization, Resources, Supervision, Funding acquisition, Project administration, Writing – review and editing

## Author ORCIDs
Isabella Tomanek http://orcid.org/0000-0001-6197-363X
Călin C Guet http://orcid.org/0000-0001-6220-2052

## Decision letter and Author response
Decision letter https://doi.org/10.7554/eLife.82240.sa1
Author response https://doi.org/10.7554/eLife.82240.sa2

## Additional files

### Supplementary files
• MDAR checklist

### Data availability
Source Data and R scripts to generate the plots shown in the Figures are uploaded as the respective source code files. Flow cytometry and Illumina sequencing data are uploaded on Dryad together with R scripts to generate the plots shown in the respective Figures (Flow cytometry data: Figure 2C, 3C-D and Figure Supplements, 4A; Illumina sequencing data: Figure 5 and Figure Supplement).

The following dataset was generated:

| Author(s) | Year | Dataset title | Dataset URL | Database and Identifier |
|---|---|---|---|---|
| Tomanek I, Guet C | 2022 | Flow cytometry YFP and CFP data and deep sequencing data of populations evolving in galactose | https://dx.doi.org/10.5061/dryad.rfj6q57ds | Dryad Digital Repository, 10.5061/dryad.rfj6q57ds |

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
