## [Editor Report]

This is an important paper that proposes a novel evolutionary mechanism by which copy-number mutations can slow down the accumulation of point mutations in populations evolving in certain environments. The authors use an evolution experiment in bacteria equipped with a clever reporter system to provide convincing evidence that this mechanism indeed operates. This paper will be of broad interest to readers in evolutionary biology and related fields.

---

## [Decision Letter]

**Decision letter after peer review:**

Thank you for submitting your article "The adaptation dynamics between copy-number and point mutations" for consideration by *eLife*. Your article has been reviewed by 3 peer reviewers, including Sergey Kryazhimskiy as Reviewing Editor and Reviewer #1, and the evaluation has been overseen by Molly Przeworski as the Senior Editor. The following individuals involved in the review of your submission have agreed to reveal their identity: Joakim Näsvall (Reviewer #2); Alejandra Rodríguez (Reviewer #3).

All reviewers agree that the paper asks an interesting question, and addresses it with a powerful reporter system. The most important issue that the reviewers identified is a lack of a convincing argument against clonal interference as an alternative explanation for the observations made in the low-demand environment. In your revision, please address this and other major criticisms listed below, and consider various suggestions for improving clarity.

Essential revisions:

(1) While your explanation involving negative epistasis makes a lot of sense, the lack of mutants carrying both duplications and point mutations in the low-demand condition could be explained by clonal interference between these types of mutations.

Please explicitly describe this alternative hypothesis in the main text and make an argument why the negative epistasis hypothesis is favored. The reviewers suggest several possibilities for making this argument.

The gold standard experiment is to construct double mutants (carrying a duplication and a point mutation that arose in the low-demand condition) and compare their competitive fitness with that of single mutants. The direct prediction of your hypothesis is that the double mutants should not be more fit than either of the single mutants. If engineering double mutants is not feasible, it may be possible instead to compare the competitive fitness (in the low-demand environments) of double mutants isolated from the high- or medium-demand environments with that of single mutants. In addition, it may also be possible to make an argument against clonal interference in the low-demand condition based on the observations made in other conditions and/or by carrying out simulations.

If a convincing argument against clonal interference cannot be made at this time, then please adjust your claims accordingly. Once the new data and/or analyses become available, you can publish them as a Research Advance article.

(2) Your main line of reasoning currently relies on the assumption that the YFP+CFP+ cells have only duplications but no promoter mutations. While this seems reasonable, we ask you to clearly spell out the argument supporting this assumption.

(3) All reviewers found the Amplification Hindrance hypothesis very interesting and useful. However, duplications are inherently unstable and are very quickly lost once selection stops favoring them. Please discuss the Amplification Hindrance hypothesis in light of this important fact.

(4) The reviewers found that your Materials and methods section lacks key pieces of information. For example, how were growth rate assays performed? How was the growth rate estimated? How were genetic reconstructions of mutations, described on LL. 472--484 and in Figure 6, carried out? This list is not exhaustive but simply illustrates the methodological gaps that the reviewers found particularly glaring. We ask you to revise the Methods section by adding enough detail so that all experiments and analyses can be reproduced.

*Reviewer #1 (Recommendations for the authors):*

LL. 241-248. I don't understand this discussion. "Despite the occurrence of yfp-only mutations in the IS- strain, increased CFP still reliably reports on increased copy-number." This sentence is self-contradictory as written. Perhaps the authors meant to add "in IS+ strains" in the second clause?

"However, the yfp-only amplification hijacks our ability to unambiguously infer ancestral copy-number from ancestral CFP fluorescence alone." This also doesn't make sense to me. Isn't the ancestral copy number known ( = 1)? Why does it need to be inferred from CFP fluorescence?

L. 253. "Mixed mutants" is a strange term. Please replace it with "double mutants" or "combination mutants". But calling populations "mixed" in the next sentence would be appropriate and helpful.

LL. 296--298. "in low galactose adaptive amplification of IS+ populations happened

rapidly with the majority of populations showing increases in CFP fluorescence during the course of the experiment". The speed has not been quantified, so I suggest making the language more precise here by dropping "rapidly". Similarly, instead of saying "the majority", it would be good to give the exact number as well as the percentage.

L. 306. "populations with clearly increased YFP levels". Which ones are these? Are these the ones labelled with red triangles in Figure 4D? If so, please say so explicitly. It would also be helpful to identify them somehow in panel 4B.

L. 310. "Sequencing of the amplified colony". Was only one colony sequenced here? I thought all YFP+ colonies were sequenced and reported in Supp Table 1. Please clarify.

L. 347. "do not occur". Point mutations presumably do occur, but they don't spread. Please correct.

L. 434-435. "Moreover, it is shifted towards higher values of YFP/CFP relative to values found for P0" Please clarify how you arrived at this claim. To say that, P0 and P0-2 must be shown on the same plot, which I do not see.

LL. 446. "with the majority of P0-2 IS+ population amplified, those few P0-2 IS+ populations that failed to evolve amplifications". Please be precise: "majority" – how many and what fraction? "Few" – how many and what fraction?

Figure 1. In panel B, it would be helpful to have error bars to see whether the decline in growth rate at higher expression levels is significant.

Figure 2:

(i) In panel A (0% galactose), please color IS- populations red as in other panels for the sake of consistency. Lines can be made more transparent to increase clarity.

(ii) In other panels, it would be helpful to show the threshold of fluorescence for calling an amplification (as a horizontal line) and then also indicate on each panel the number and fraction of populations that have acquired an amplification by the end of the experiment.

(iii) In panel B, I would suggest doing the same thing as above, i.e., showing the number (and fraction) of populations in each class e.g., YFP+CFP+ (18). Also, please show grid lines and the diagonal so that it is possible to compare data across panels.

Figure 3: It is currently impossible to associate each panel in C with a specific point in panel A because multiple points have the same color. One possible way to resolve this ambiguity is to draw the trajectories in A as lines but only show one time point (the one that is shown as a panel in C) with a circle.

Figure 4: I have several issues with panel A.

(i) How are the cell counts normalized and why?

(ii) I think the authors should consider splitting the CFP vs YFP plot into two, one showing population B1 and the other showing population B3. Right now, B1 completely obscures B3. I also suggest somehow highlighting the three subpopulations that the authors refer to in the text on LL. 308-309.

(iii) I don't understand whether the right-most panel and the middle set of panels show the same data. Specifically, on the CFP vs YFP plot, it is clear that population B1 has a subpopulation with log(CFP) ~ 6 and log(YFP) ~ 6. On the YFP plot, there is a visible tail around the same value, but the CFP plot does not seem to have a corresponding peak. Is this a plotting error? If not, perhaps the issue is with the normalization, or maybe the cell counts need to be plotted on the log scale.

(iv) It seems important for the authors' argument to show that ALL populations evolved in 0.01% show the same pattern as B1 and B3. I would suggest showing the YFP vs CFP plot for every population as a supplementary figure.

(v) Panel B. It is nice to show the raw data, but I think the exposition would improve if this panel was in the supplement. The reason is that differences in YFP fluorescence values are very difficult to discern by eye, and the same data (as far as I understood) are anyway presented in a more discernible way in panel D.

Figure 5:

(i) In the caption and in the text (l. 369) the authors refer to the "number of reads", but I think in all case they mean "fraction of reads". I am also a bit confused what the authors mean by "normalized to the number of reads with ancestral P0". Do they show the actual fraction of reads carrying the respective mutations in each set of population or is it something else?

(ii) I would suggest adding the position number to the labels of point mutations, e.g., T30A, C37T

Figure 6:

(i) please add a legend to panel A, as in Figure 2A.

(ii) I am very confused about panel C. If I understand it correctly, it shows that >20 colonies sampled from IS- populations are labelled as YFP+CFP+. Based on what data was this assignment made? Panel A shows that there are apparently no IS- populations that have the YFP+CFP+ phenotype, and we don't expect them to happen at such high rates in the IS- populations. Please clarify.

(iii) In panel E, mutants are labeled A11, F2, etc., but I don't seem to find a correspondence between these labels and the specific mutations that have been reconstructed. Please clarify.

(iv) By the same token, I don't see a description of how these mutants were constructed.

Figure 7B: The double-mutant is missing from this panel. I understand that the authors have not observed such double-mutants in low-demand environments, but I think it would be helpful to show it for the purposes of clarifying their hypothesis.

*Reviewer #2 (Recommendations for the authors):*

I find the present manuscript acceptable for publication but think the authors could consider adding a few points where it's suitable:

1. Could the authors discuss or comment on the "possible weakness" mentioned in the public review?

2. The evolutionary dynamics at different galactose concentrations are shown in Supplementary Figure 2, but there might be more to be learned by looking at the exact conditions of the evolution experiments (M9+casaminoacids+various concentrations of galactose): what are the growth dynamics of the ancestral strain and the evolved strains? Is the main evolutionary advantage on the maximum growth rate, final population size, or somewhere else (or a combination of several parameters)? If data on this is not already available I think it should be a quick and simple experiment to add (note that I do not think this is critical for publication, but that it could potentially add some additional value to the discussion).

3. Could selection at slightly different conditions (smaller/larger populations, smaller/larger bottlenecks) or with fluctuating selection pressures affect the outcome? Could variations in the conditions used produce a different skew of duplications vs. promoter mutations?

I enjoyed reading the manuscript and hope to see it published soon!

*Reviewer #3 (Recommendations for the authors):*

Overall, I think is an exciting study and the researcher was rigorously conducted and well presented.

---

## [Author Response]

Essential revisions:(1) While your explanation involving negative epistasis makes a lot of sense, the lack of mutants carrying both duplications and point mutations in the low-demand condition could be explained by clonal interference between these types of mutations.Please explicitly describe this alternative hypothesis in the main text and make an argument why the negative epistasis hypothesis is favored. The reviewers suggest several possibilities for making this argument.The gold standard experiment is to construct double mutants (carrying a duplication and a point mutation that arose in the low-demand condition) and compare their competitive fitness with that of single mutants. The direct prediction of your hypothesis is that the double mutants should not be more fit than either of the single mutants. If engineering double mutants is not feasible, it may be possible instead to compare the competitive fitness (in the low-demand environments) of double mutants isolated from the high- or medium-demand environments with that of single mutants. In addition, it may also be possible to make an argument against clonal interference in the low-demand condition based on the observations made in other conditions and/or by carrying out simulations.If a convincing argument against clonal interference cannot be made at this time, then please adjust your claims accordingly. Once the new data and/or analyses become available, you can publish them as a Research Advance article.

We thank the reviewers for the suggested experiment, which we agree would unequivocally distinguish between epistasis and clonal interference. We will actually do this experiment and we would like to take the opportunity to publish it as a Research Advance article. For the current revision however, we could not do this, as the first author is already in Oxford doing her postdoctoral work and is tied up by family obligations as well (two small kids) from travelling to perform this work in Vienna, at the present time.

Therefore, we chose to address this criticism in two ways: (i) We explicitly discuss clonal interference (CI) in the main text while at the same time we weaken our epistasis claims to entertain the possibility of CI, as we are not showing the presence of epistasis directly with experiments. (ii) We also present arguments (in this response but also in the main text) of why we think epistasis is the more likely explanation. Please see the final section of the Results section, starting at line 551 (together with the newly added Figure 7 —figure supplement 1).

Clonal interference (CI) is the effect of the competition between lineages arising from different beneficial mutations in a population without recombination (i.e. asexually reproducing). When the product of population size and mutation rate (N*µ) is smaller than one (N*µ<1), mutations fix one by one, whereas if N*µ exceeds one significantly (N*µ>>1), mutations compete, thereby slowing down the fixation of each other (Gerrish and Lenski, 1998; Park and Krug, 2007).

At the first glance, the fact that significantly more replicate populations become amplified in the lowest galactose concentration suggests that clonal interference could play a role in this environment. All else being equal, we expect CI to occur for larger population sizes or at higher mutation rates. However, in the absence of strong *galK* expression (i.e. expression levels of.*galK* with ancestral P0) the maximal OD_600_ (“yield”) as a proxy for population size is similar between populations in the different galactose environments (please see new Figure 7 —figure supplement 1a). The reason for this is that in the absence of *galK* expression the cells essentially only metabolize the casaminoacid component of the evolution medium, which is the same across all galactose environments.

At the same time, there is no reason to assume that the mutation rate should differ between the different galactose concentrations. Instead, the observation that amplifications evolve faster in the low galactose environment can be explained by differences in the expression-growth relation for *galK* (“fitness landscape”, Figure 1b) between the different environments. Specifically, it appears as though the environments differ in their adaptive benefit of increasing *galK* expression starting from a low level (at low levels of inducer arabinose in Figure 1b). This would imply that acquiring a duplication of *galK* incurs little fitness benefit in 1% galactose (the fitness landscape exhibits a plateau for low expression values), while the same mutation incurs a bigger benefit in 0.01% galactose (the fitness landscape is slightly steeper when going from one copy of *galK* to two copies of *galK*). Because we can with relative confidence rule out both very significant differences in the population size and differences in the duplication rate between environments, CI is an unlikely explanation for the fact that we see mutually exclusive mutations in low galactose, but not in higher galactose concentrations.

There is one additional argument to be made in favour of epistasis: In our (new) Figure 7 —figure supplement 1b-d, the growth of amplified strains (blue) that have evolved in the three different Gal concentrations for seven days, respectively, is compared to the growth of two different promoter mutants (double mutant, and slightly weaker single mutant, both in green) (for convenience, also see Figure 7 —figure supplement 1b-d). Intriguingly, the difference between both yield and growth rate of amplified and promoter mutant strains is bigger in both high galactose environments (1% and 0.1%) than in low galactose (0.01%). The amplified strains, which have evolved in low galactose are growing comparably well to the promoter mutant. One of the amplified strains (blue continuous lines) even reaches higher population densities (yields) than either promoter mutants. This suggests that amplification can in fact provide sufficient expression in low galactose, while a higher expression level is required in high galactose. These results are in agreement with the epistasis hypothesis, namely that additional point mutations would be beneficial in the higher galactose concentrations, but that they would yield little or no fitness benefit in low galactose (=resulting in negative epistasis).

Finally, a second line of evidence argues against clonal interference playing a role in making mutations mutually exclusive in low galactose. CI essentially means that many adaptive mutations arise simultaneously but in different individuals. Hence, beneficial mutations are not only competing against the ancestral (low fitness) allele, but against all other beneficial alleles. Its fixation is therefore slowed down. However, even in low galactose, despite the rapid amplification of most populations, a significant fraction of cells is still ancestral by the end of the evolution experiment (see new Figure 4 —figure supplement 1, Figure 4a). This suggests that the population is not yet “saturated” with competing mutations. Extending this argument, in the case of CI we would expect to find at least small subpopulations of combination mutants when monitoring populations both, on the single-cell level (Figure 4 —figure supplement 1, Figure 4a) or single-colony level (essentially corresponding to single cells; e.g. Figure 4C).

The above listed arguments, together with the fact that the maximal copy-number differs significantly for evolved populations in the three environments (Figure 2 – Figures Supplement 1b) strongly suggests (even in the absence of direct experimental evidence) that combination mutants are absent in the low demand condition because additional point mutations are obsolete (i.e. negative epistasis).

We made the following changes in the main text to address the point about CI versus epistasis:

– We adjusted the abstract by removing the claim of epistasis (and pointing out the high frequency of amplifications, which is underlying amplification hindrance irrespective of whether it occurs through negative epistasis or clonal interference): line 23.

– We adjusted line 109 in the Introduction: “However, if both, copy-number and point mutations are adaptive (Gruber *et al.*, 2012), they also have the potential to interact epistatically or due to clonal interference.”

– We added a new section to the final part of the Results, starting at line 551 (together with the newly added Figure 7 —figure supplement 1).

– We adjusted the discussion in line 626 (“most likely epistasis”)

(2) Your main line of reasoning currently relies on the assumption that the YFP+CFP+ cells have only duplications but no promoter mutations. While this seems reasonable, we ask you to clearly spell out the argument supporting this assumption.

We agree that this is an important point which we have not made sufficiently explicit (in part because YFP+CFP+ cells having only copy-number mutations has been a repeated observation we have had in past studies (Steinrueck and Guet, 2017; Tomanek *et al.*, 2020)).

We adjusted the manuscript in two places to make this point clearer and also support it with additional data:

(i) Using data from previous experiments, we added a new Figure 1 —figure supplement 1, which shows the correlation between YFP and CFP and copy-number as measured by qPCR. In these plots, a promoter mutant ((-30T>A and -37C>T)) is an obvious outlier in the levels of YFP. This is mentioned in line 153.

(ii) In Figure 6C we summarise the sequencing result of (among other things) 21 colonies from the amplified fraction (i.e. with a correlated increase in both fluorophores (“YFP+CFP+”)), which all harbour an ancestral P0-2 sequence. We point this out more clearly in the text now (line 520).

(3) All reviewers found the Amplification Hindrance hypothesis very interesting and useful. However, duplications are inherently unstable and are very quickly lost once selection stops favoring them. Please discuss the Amplification Hindrance hypothesis in light of this important fact.

We agree with this point and want to emphasise that we are aware of the unstable nature of amplifications, which we have studied previously under regimes of fluctuating selection (Tomanek *et al.*, 2020).

In our previous study, we evolved populations with amplifications in P0-*galK* in environments alternatingly selecting for and against *galK* expression. Under our experimental conditions, copy-number polymorphisms accumulate rapidly and provide an ample diversity of *galK* expression levels for selection to pick from. Most likely because evolutionary dynamics were dominated by strong selection and high rates of duplication/deletion, in that study we failed to recover any point mutations in P0 in populations evolved under our fluctuating conditions.

In fact, it is plausible that under fluctuating selection amplification hindrance may be even stronger than under continuous selection. The reason for this is that promoter point mutations leading to constitutive expression might be slightly costly when expression is not selected for. Unlike the rate of reversal of point mutations, amplifications have a very high rate of reversal that allows amplified populations to rapidly attain ancestral (low) expression levels, only to re-appear once selection for increased gene expression resumes.

That being said, we agree with the general point that the study uses simplified environments. Therefore, we would be very interested in whether the phenomenon described here actually occurs in natural/clinical settings (e.g. antibiotic resistance genes that are known to be amplified and causing heteroresistance) and are more conserved than comparable resistance genes, which are not amplified.

In our present manuscript, we touched on this topic in the Discussion section (see below). We now added the following statements to more explicitly discuss fluctuating selection (line 694).

“If adaptation by amplification is the fastest and sufficient, other, less frequent, mutations may not have a chance to compete. While our evolution experiments were conducted under continuous selection, natural environments may provide regimes of fluctuating selection. Due to the pleiotropic cost often associated with copy-number increases as well as their high rate of deletion, adaptive amplification returns to the ancestral single copy state in the absence of selection (Andersson and Hughes, 2009; Reams *et al.*, 2010). This means that once the selective benefit of the transient adaptation is gone, no change at the level of genomic DNA remains (Roth *et al.*, 1996). Therefore, the idea that gene amplifications act as a transient “regulatory state” rather than a mutation (Roth *et al.*, 1996; Tomanek *et al.*, 2020) can be extended by an implication found here, namely that amplifications could effectively act as buffer against long-lasting point mutations. In this view, amplification could repeatedly provide rapid adaptation to selection for increased gene expression, but collapse back to the single copy ancestral state once selection has subsided and yet hinder sequence divergence each time it does so. Thus, on sufficiently long time-scales, the transient dynamics that play out before the fixation of mutations may ultimately shape entire genomes”

(4) The reviewers found that your Materials and methods section lacks key pieces of information. For example, how were growth rate assays performed? How was the growth rate estimated? How were genetic reconstructions of mutations, described on LL. 472--484 and in Figure 6, carried out? This list is not exhaustive but simply illustrates the methodological gaps that the reviewers found particularly glaring. We ask you to revise the Methods section by adding enough detail so that all experiments and analyses can be reproduced.

We thank the reviewers for pointing out this issue. We added the following new sections to Methods:

(i) OD_600_ and fluorescence measurements in liquid cultures and calculation of growth rate (line 824)

(ii) Reconstitution of P0-2 mutants in the ancestral strain by oligo-recombineering (line 768)

Reviewer #1 (Recommendations for the authors):LL. 241-248. I don't understand this discussion. "Despite the occurrence of yfp-only mutations in the IS- strain, increased CFP still reliably reports on increased copy-number." This sentence is self-contradictory as written. Perhaps the authors meant to add "in IS+ strains" in the second clause?

We agree that this sentence (and the following mentioned below) was confusing as stated. To make it clearer we now elaborate more on the issue. In fact, increased CFP still reports on increased copy-number in both IS+ and IS- strain (although rare in the latter due to its lowered rate of (IS-dependent) amplification).

We replaced the sentence in line 258 with the following paragraph:

“While increased CFP still reliably reports on increased copy-number, the yfp-only amplification hijacks our ability to unambiguously infer ancestral copy-number from ancestral CFP fluorescence alone. As increasing CFP itself bears no adaptive benefit, populations with increased CFP must carry amplifications that also include *galK*. In contrast, ancestral copy-number can only be confirmed by qPCR. The fact that some populations carry IS-independent yfp-only amplifications implies that our system of fluorescence reporters will yield a slight underestimate of the number of amplified populations both in the IS+ and IS- strain. However, we were ultimately interested in the divergence of promoter sequences, and going forward relied on sequencing to unambiguously determine the presence of adaptive promoter mutations.“

"However, the yfp-only amplification hijacks our ability to unambiguously infer ancestral copy-number from ancestral CFP fluorescence alone." This also doesn't make sense to me. Isn't the ancestral copy number known (=1)? Why does it need to be inferred from CFP fluorescence?

We corrected the issue together with the preceding sentence. Please see above paragraph.

L. 253. "Mixed mutants" is a strange term. Please replace it with "double mutants" or "combination mutants". But calling populations "mixed" in the next sentence would be appropriate and helpful.

We changed “mixed mutants” to “double mutants”.

LL. 296--298. "in low galactose adaptive amplification of IS+ populations happened rapidly with the majority of populations showing increases in CFP fluorescence during the course of the experiment". The speed has not been quantified, so I suggest making the language more precise here by dropping "rapidly". Similarly, instead of saying "the majority", it would be good to give the exact number as well as the percentage.

The numbers were unfortunately buried in the captions of Supplementary Figure 1A (now Figure 1 —figure supplement 1a). We now added them to the main text (pointing also to the Supplement for our definition of what constitutes an “amplified” population). We changed the wording to “more rapidly” (i.e. a relative term) as we found it important to draw the readers´ attention to the difference between the environments (line 319).

L. 306. "populations with clearly increased YFP levels". Which ones are these? Are these the ones labelled with red triangles in Figure 4D? If so, please say so explicitly. It would also be helpful to identify them somehow in panel 4B.

Yes, indeed. We added “(Figure 4d – red triangles)” to clarify.

L. 310. "Sequencing of the amplified colony". Was only one colony sequenced here? I thought all YFP+ colonies were sequenced and reported in Supp Table 1. Please clarify.

Yes, indeed, we sequenced all YFP+ colonies (corresponding to the red triangles in Figure 4D and reported in Table 1). The reviewer is right, as the way we were reporting our results was confusing. We now make it clearer by first reporting on one specific population (B1, the one in Figure 4a-b) for which we show the agar streak to illustrate our procedure for sequencing and point out the obvious difference in phenotype that can be noticed “by eye” (Figure 4c). We then explain that we sequenced the remaining YFP+ (red triangle) populations in the same manner.

L. 347. "do not occur". Point mutations presumably do occur, but they don't spread. Please correct.

We corrected the wording to: “If copy-number mutations are more frequent than point mutations and their combination does not spread to observable frequencies in the low demand environment…” (line 379)

L. 434-435. "Moreover, it is shifted towards higher values of YFP/CFP relative to values found for P0" Please clarify how you arrived at this claim. To say that, P0 and P0-2 must be shown on the same plot, which I do not see.

In order to clarify, we added Figure 6 —figure supplement 1b, where we plot YFP/OD600 against CFP/OD600 for both IS+ populations of P0 and P0-2 in one plot.

LL. 446. "with the majority of P0-2 IS+ population amplified, those few P0-2 IS+ populations that failed to evolve amplifications". Please be precise: "majority" – how many and what fraction? "Few" – how many and what fraction?

We added this information (“majority”: 88/96 and “few”: 6/96) to the main text (line 487). We also added the threshold of considering a population amplified (mean of ancestral CFP + 4 SD of ancestral CFP) as a green line in Figure 6 —figure supplement 1A.

Figure 1. In panel B, it would be helpful to have error bars to see whether the decline in growth rate at higher expression levels is significant.

The plot shown in Figure 1 panel B is a single replicate. We have repeated the experiment twice three years later and it qualitatively produced similar results, most importantly the fact that growth rate increases with increasing *galK* expression, but levels off at different induction levels for different galactose concentrations (please see Author response image 1). The differences between replicate experiments are probably mainly due to batch effects in the sugar- and casaminoacids stocks used for preparing the different media.

As the reviewer noted, it seems there is indeed a growth rate reduction in 1% arabinose (in the control, i.e. in the absence of galactose), but this is likely an artifact of the strain we are using rather than the cost of high *galK* expression itself. We used strain background BW27784 for our chromosomal *pBAD-galK* construct. BW27784 has the advantage of allowing an almost linear induction behaviour for the arabinose promoter (pBAD) as opposed to the none-to-all behaviour typical of most catabolic promoters (Khlebnikov et al., 2001). To this end, it carries a constitutive arabinose import system and a deletion of the arabinose catabolic genes. Therefore, BW27784 is not able to catabolize arabinose, such that high concentrations are probably toxic. Importantly, our evolution experiments we used an MG1655 strain background (and no arabinose), so we don’t expect high levels of *galK* expression to be toxic. Indeed, we do not see a cost of *galK* expression even in the absence of galactose when comparing strains with very low levels of *galK* expression (ancestral P0) and strains with high *galK* expression (P0 double promoter mutant) such as in Figure 2 —figure supplement 2b.

Moreover, we are confident that the data in Figure 1b capture the most important aspects of the *galK* expression-growth relation in the different environments, as the level of amplification (#of copies) evolving in our evolution experiment increases with increasing galactose concentration (Figure 2 —figure supplement 1B).

**Author response image 1. sa2fig1:** Replicates of the Experiment shown in Figure 1b, where growth rate is plotted against arabinose concentration used to induce *galK* expression for different galactose concentrations (line colors, see legend). Left panels show all tested galactose concentrations, while right panels show the relevant galactose concentrations used in the experiments of the manuscript.

Figure 2:(i) In panel A (0% galactose), please color IS- populations red as in other panels for the sake of consistency. Lines can be made more transparent to increase clarity.(ii) In other panels, it would be helpful to show the threshold of fluorescence for calling an amplification (as a horizontal line) and then also indicate on each panel the number and fraction of populations that have acquired an amplification by the end of the experiment.

We are showing the threshold of fluorescence for determining an amplification in Figure 2 —figure supplement 1A.

We report the numbers in the main text and added them to the figure as well.

(iii) In panel B, I would suggest doing the same thing as above, i.e., showing the number (and fraction) of populations in each class e.g., YFP+CFP+ (18). Also, please show grid lines and the diagonal so that it is possible to compare data across panels.

We added grid lines to the plots and added the numbers to the figure panel.

Figure 3: It is currently impossible to associate each panel in C with a specific point in panel A because multiple points have the same color. One possible way to resolve this ambiguity is to draw the trajectories in A as lines but only show one time point (the one that is shown as a panel in C) with a circle.

We drew evolutionary trajectories in color and added the last time point in addition to the lines, as suggested by the reviewer.

Figure 4: I have several issues with panel A.(i) How are the cell counts normalized and why?

For this plot we use the geom_density() function of the R package ggplot2 to draw a kernel density estimate, a smoothed version of the underlying histogram, as this is often used for flow cytometry data in the field. The default plot used by us scales the data such that the area under the curve equals 1. We now replaced “histogram” in the caption with “density plot”.

(ii) I think the authors should consider splitting the CFP vs YFP plot into two, one showing population B1 and the other showing population B3. Right now, B1 completely obscures B3. I also suggest somehow highlighting the three subpopulations that the authors refer to in the text on LL. 308-309.

We split the plots into two as suggested and added arrows to highlight subpopulations.

(iii) I don't understand whether the right-most panel and the middle set of panels show the same data. Specifically, on the CFP vs YFP plot, it is clear that population B1 has a subpopulation with log(CFP) ~ 6 and log(YFP) ~ 6. On the YFP plot, there is a visible tail around the same value, but the CFP plot does not seem to have a corresponding peak. Is this a plotting error? If not, perhaps the issue is with the normalization, or maybe the cell counts need to be plotted on the log scale.

We agree that the plotting has not been ideal. The data is indeed the same. Cell numbers of the amplified subpopulation in B1 are small (this is hard to judge from the YFP-CFP plot as it was shown). The amplified sub-peak is actually there in the histogram but basically not visible in the small plot we show (please see histograms in a log scale, author response image 2). Therefore, we now stretched the plot along the y-axis and added a small arrow to highlight the rather tiny peak.

**Author response image 2. sa2fig2:** Histogram of population B1 CFP fluorescence intensity [a.u.] plotted on a log scale to visualize the small amplified subpopulation (black=day 4, blue=day 8, purple = day 12).

(iv) It seems important for the authors' argument to show that ALL populations evolved in 0.01% show the same pattern as B1 and B3. I would suggest showing the YFP vs CFP plot for every population as a supplementary figure.

We now added this data as new Figure 4 —figure supplement 1a-b.

(v) Panel B. It is nice to show the raw data, but I think the exposition would improve if this panel was in the supplement. The reason is that differences in YFP fluorescence values are very difficult to discern by eye, and the same data (as far as I understood) are anyway presented in a more discernible way in panel D.

We added the data as a new Figure 4 —figure supplement 2a, since (as correctly pointed out) it is quantified anyways in panel D.

Figure 5:(i) In the caption and in the text (l. 369) the authors refer to the "number of reads", but I think in all case they mean "fraction of reads". I am also a bit confused what the authors mean by "normalized to the number of reads with ancestral P0". Do they show the actual fraction of reads carrying the respective mutations in each set of population or is it something else?

Our wording here was indeed misleading. We used “numbers” with the intent to describe in simple words how our algorithm works, but we should have instead said fractions from the beginning of the paragraph, to avoid confusion. We fixed this issue now. By “Normalized to the number of reads with ancestral p0” we meant that we divided #reads_with_mutations/#reads_without_mutations. We added this as an explicit statement in line 397 and also corrected a similar issue in the methods (paragraphs after line 894).

(ii) I would suggest adding the position number to the labels of point mutations, e.g., T30A, C37T

We added the position numbers in the plot legends as suggested.

Figure 6:(i) please add a legend to panel A, as in Figure 2A.

Thank you for pointing that out, we added now both the fraction names and the respective numbers.

(ii) I am very confused about panel C. If I understand it correctly, it shows that >20 colonies sampled from IS- populations are labelled as YFP+CFP+. Based on what data was this assignment made? Panel A shows that there are apparently no IS- populations that have the YFP+CFP+ phenotype, and we don't expect them to happen at such high rates in the IS- populations. Please clarify.

Thank you for spotting this mistake. We unintentionally seemed to have swapped the labels of “YFP+CFP+” and “ancestral” while preparing the figure from the original R plots. Indeed, IS- just has a single CFP+YFP+ population. We actually sequenced 21 ancestral colonies instead. Similarly, for the IS+ strain the numbers of sequenced colonies are swapped: we sequenced 17 YFP+CFP+ and 14 ancestral ones (not the other way around). We now corrected this mistake so that the table makes sense again (and is in agreement with the numbers we describe in the text).

(iii) In panel E, mutants are labeled A11, F2, etc., but I don't seem to find a correspondence between these labels and the specific mutations that have been reconstructed. Please clarify.

Thank you for pointing that out. In one case we had added the population name, in the other the kind of mutation (both types of info are combined in the Table 2). We corrected the issue now so the labels are corresponding.

(iv) By the same token, I don't see a description of how these mutants were constructed.

We added this missing information in the methods section named: Reconstitution of P0-2 mutants by oligo-recombineering.

Figure 7B: The double-mutant is missing from this panel. I understand that the authors have not observed such double-mutants in low-demand environments, but I think it would be helpful to show it for the purposes of clarifying their hypothesis.

We followed the suggestion and added the combination mutant to the scheme. We connected this mutant with the other ones by faint horizontal lines indicating neutral evolution (described in the captions).

Reviewer #2 (Recommendations for the authors):I find the present manuscript acceptable for publication but think the authors could consider adding a few points where it's suitable:1. Could the authors discuss or comment on the "possible weakness" mentioned in the public review?

We discuss the issue about epistasis versus clonal interference in our reply here (please see the section starting on page 1) and made several adjustments to the text (please see explanations above).

2. The evolutionary dynamics at different galactose concentrations are shown in Supplementary Figure 2, but there might be more to be learned by looking at the exact conditions of the evolution experiments (M9+casaminoacids+various concentrations of galactose): what are the growth dynamics of the ancestral strain and the evolved strains? Is the main evolutionary advantage on the maximum growth rate, final population size, or somewhere else (or a combination of several parameters)? If data on this is not already available I think it should be a quick and simple experiment to add (note that I do not think this is critical for publication, but that it could potentially add some additional value to the discussion).

This is an interesting point. The data behind Figure 1b (where we induced expression of *galK* along a gradient of arabinose and measured growth rates in the different galactose concentrations) hints at the fact that both growth rate and population size (yield) will be increased with increasing either *galK* expression or galactose concentration (at a given expression level). The data also suggests that when *galK* is expressed a low level, low galactose concentrations result in higher growth rate. (As mentioned above, we suspect that this is the reason behind the steeper fitness peak for low galactose and the reason why amplifications evolve there more rapidly than in the higher galactose concentrations). Please see Author response image 3.

**Author response image 3. sa2fig3:** Example growth curves in M9 evolution medium and a 2D gradient of galactose and arabinose (as an inducer of *galK* expression). These measurements are the basis for Figure 1b. The top left panel shows growth in 1% glycerol (control) instead of galactose.

The idea that both, yield and growth rate are increased by increasing *galK* expression (irrespective of the nature of the underlying mutation) is corroborated by the growth curves of the different evolved strains in the different galactose concentrations. Specifically, we compared the growth of six different amplified strains, where two evolved in one of the three different galactose concentrations, respectively, to two promoter mutants (the double mutant “H5” and a single mutant isolated in a previous study). While we do not have data on how the ancestral strain compares in the same environment, the data from the different mutants suggests that populations gain both, growth rate and yield increases with increasing galactose concentration (please see new Figure 7 —figure supplement 1; for convenience the figure is also plotted within Author response image 3 where we discuss it in the context of clonal interference/epistasis – Figure 2).

3. Could selection at slightly different conditions (smaller/larger populations, smaller/larger bottlenecks) or with fluctuating selection pressures affect the outcome? Could variations in the conditions used produce a different skew of duplications vs. promoter mutations?I enjoyed reading the manuscript and hope to see it published soon!

Intuitively, we think that slightly different population sizes/ bottlenecks should not matter too much for the experiments shown. We gather this information from some preliminary experiments where we used different dilution factors, and one experiment where we also used a 10-fold bigger population size.

However, the rather big difference in outcome between the different environments seems to suggest that the distribution of mutations should quite sensitively depend on the galactose concentration, especially for lower galactose concentrations. We would speculate that the fitness landscape (shown in Figure 1b) if resolved at a finer scale (i.e. plotting growth rate along a series of very small increases in gene expression), might show an initial plateau for the high galactose concentrations, but not for the low one. We suspect that metabolic/biochemical constraints differ in the different galactose concentrations and the strength of selection will not simply be a linear function of the galactose concentration.